# MULocBench: Beyond Code Benchmark for Multi-Type Software Issue Localization

## Abstract

Accurate project localization (e.g., files and functions) for issue resolution is a critical first step in software maintenance. However, existing benchmarks for issue localization, such as SWE-Bench and LocBench, are limited. They focus predominantly on pull-request issues and code locations, ignoring other evidence and non-code files such as commits, comments, configurations, and documentation. To address this gap, we introduce MULocBench, a comprehensive dataset of 1,100 issues from 46 popular GitHub Python projects. Comparing with existing benchmarks, MULocBench offers greater diversity in issue types, root causes, location scopes, and file types, providing a more realistic testbed for evaluation. Using this benchmark, we assess the performance of state-of-the-art localization methods and five LLM-based prompting strategies. Our results reveal significant limitations in current techniques: even at the file level, performance metrics (Acc@5) remain below 40%. This underscores the challenge of generalizing to realistic, multi-faceted issue resolution. To enable future research on project localization for issue resolution, we publicly release MULocBench at https://huggingface.co/datasets/somethingone/MULocBench.

## 1 Introduction

Modern software projects are inherently complex. They often consist of thousands of files spanning code, configurations, tests, and documentation. The complexity making developers routinely encounter a wide spectrum of issues, ranging from runtime failures and unexpected results to enhancement requests and usage questions. A prerequisite for resolving these issues is to accurately identify the locations, such as the relevant files and functions.

Existing benchmarks have advanced research on issue localization. SWE-Bench Jimenez et al. collects 2,294 issues with pull requests from 12 Python projects, primarily targeting bug fixing. To encourage adoption, it releases SWE-bench Lite, a subset of 300 instances. Most Recently, LocBench Chen et al. (2025) expands the scope to 560 issues with more diverse types. Despite these efforts, existing benchmarks remain limited. They focus narrowly on code and rely solely on pull requests as evidence of resolution, overlooking other resolution signals and artifact types. In practice, issues may be resolved not only through pull requests but also via commits and even explicit comments. Moreover, project locations often involve files beyond source code, such as configurations, documentation, and third-party libraries.

To better reflect these real-world scenarios, we present MULocBench, a dataset of 1,100 issues drawn from 46 popular GitHub Python projects. Each issue is linked to a pull request, commit, or resolution-confirming comment, and includes detailed location information—project name, file path, class name (if applicable), function name (if applicable), and line numbers (if applicable). MULocBench offers broader and more balanced coverage across issue types, root causes, location scopes, and file types, providing a richer foundation for evaluating localization approaches.

To evaluate how well existing techniques support realistic issue localization, we conduct two experiments based on MULocBench. We first investigate the effectiveness of state-of-the-art approaches( retrieval-based, procedure-based, and agent-based methods) on locating identification within in-project Python code. LocAgent and OpenHands achieve the best and comparable results, with Acc@5 reaching at most 35.2%, 28.5%, and 25.0% at the file, class, and function levels.

Figure 1: MULocBench Construction Overview

The result is substantially lower than the 60%+ Acc@5 consistently observed across all three levels on SWE-Bench Lite and LocBench. We then further examine whether large language models (LLMs) can go further, identifying locations not only in code files and in-project files. We design five LLM-based prompting strategies with varying levels of contextual support, from closed-book to pipeline(Location-Hint guided) localization. The best performance with Location-Hint guided reaches only around 35.0%, 24.5%, and 21.0% Acc@5 at the file, class, and function levels.

In summary, the main contributions are: (1) MULocBench, a project location benchmark with 1,100 issues across diverse issues and locations. (2) A systematic categorization of issue types and causes, and location scopes and types. (3) A comprehensive comparison with state-of-art location benchmarks. (4) A systematic evaluation highlighting the limitations of current localization techniques and LLMs in project localization for issue resolution.

## 2    Benchmark Construction

Project developers encounter numerous and diverse issues in the process of software development. Location identification is a prerequisite for resolving software issues. To build a benchmark for location information, we draw on GitHub Issues. The benchmark construction is a three-stage pipleine, as shown in Figure 1. First, we select the top-50 starred Python repositories and randomly sample 200 issues per project to balance diversity and manual effort. Then, we retain only closed, resolved issues with reliable location via merged PRs, commits, or explicit comments; for issues without PRs/commits, comments must clearly state the location. Finally, we extract locations (project name, file path, class, function, line) automatically from PR/commit patches or manually from resolution comments. The details are as follows:

**(1) Issue selection:** The process begins by specifying a project set, then selecting issues within those projects. We focus on the top 50 most-starred GitHub Python projects, which typically have large communities and rich issue discussions, making them ideal for constructing representative and diverse benchmarks. Python is chosen for its top-ranked popularity and broad domain applicability, which often leads to diverse project-related issues. For each project, we randomly sample 200 issues to balance manual effort with question diversity.

**(2) Issue filtering:** Reliable location information is critical for our study, but many issues lack such information due to being open, unresolved, or having ambiguous descriptions. To obtain reliable location data, we apply three filtering conditions: selecting closed issues, retaining those with confirmed resolutions, and keeping only issues with clear location information. After filtering, there are 1,100 issues from 46 projects. Among them, 716 have pull requests, 63 have commits, and 321 have resolution-related comments. The details of filtering stage are in Appendix A.1.

**(3) Location extraction:** Location information includes *project name*, *file path*, *class name*, *function name*, and *line numbers*. For issues linked to pull requests or commits, location information is automatically extracted by parsing the corresponding patch files. For issues without linked pull requests or commits, we manually extract location information by analyzing the resolution-related comments. All manual extracted location have been confirmed by authors. For example, consider the comment: just changed line *v2.VideoCapture(camera_index)* in ui.py. By inspecting the repository, we identify the location as follows: the project name is `hacksider/Deep-Live-Cam`, the file path is `modules/ui.py`, the class name is empty (as the line is not within a class), the function name is `render_video_preview`, and the line number is 203. In another example, the location refers to a configuration file rather than source code. The comment states: "It works with Python 3.10.12 when numpy version is changed to numpy==1.22.0 in requirements.txt." Here, the file path is `requirements.txt`, the class name and function name are empty, and the line number is 4.

## 3    Empirical Analysis

After constructing MULocBench, a project-level, location-centric benchmark, we conduct an empirical study of issue reports and their resolution locations. Prior work Jimenez et al.; Chen et al.

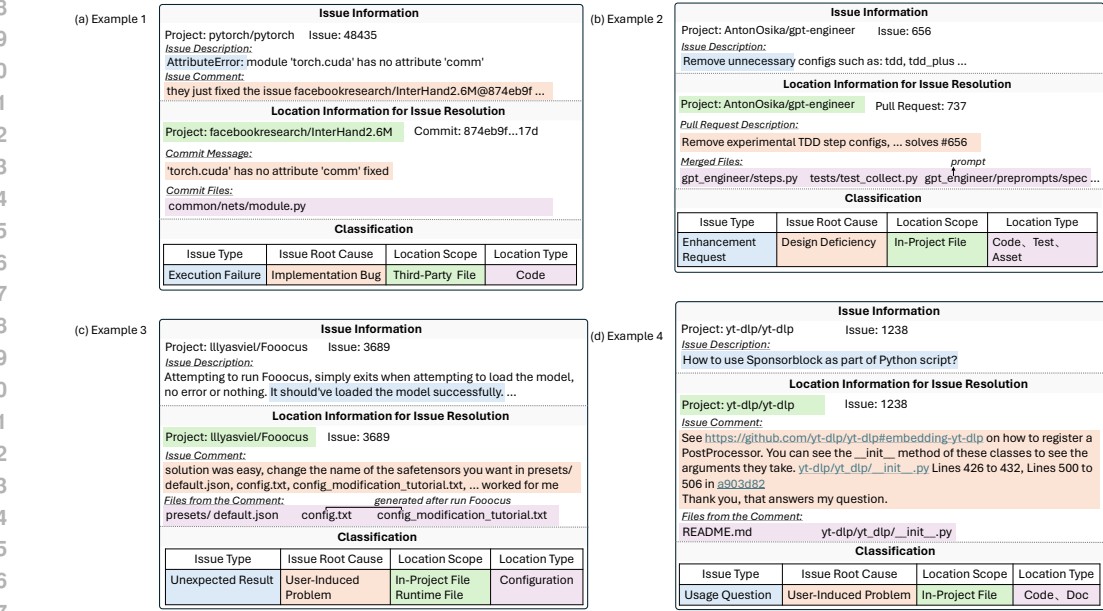

Figure 2: Four issues illustrating different issue types, reasons, location scopes and types.

(2025) has mainly curated issue–patch pairs and focused on bug fixing, but offered little analysis of issue types, causes, or affected files. Such analysis is crucial for understanding benchmark characteristics and enabling meaningful comparisons. Illustrative examples of issue types, root causes, location scopes, and location types are shown in Figure 2. The analysis methods and definitions of issue types, reasons, and location scopes and types detailed in Appendix A.4.1 and Appendix A.4.2.

**Issue Types** capture the problems developers raise in issue reports, such as execution errors or usage questions. We classify issues into four categories: Execution Failures (39.9%), Unexpected Results (23.7%), Enhancement Requests (25.1%), and Usage Questions (11.3%). For example, in panel (d) of Figure 2, a developer asks how to use `Sponsorblock` as part of a Python script, which is classified as a Usage Question because it reflects a request for guidance.

**Root Causes of Issues** capture why problems arise. We categorize them into Implementation Bugs (34.5%), Design Deficiencies (36.3%), and User-Induced Problems (29.2%). For example, in panel (c) of Figure 2, developers fail to load a model in `Fooocus` until they correct their own configuration settings. It represents a User-Induced Problem, as the issue stems from incorrect usage rather than flaws in the project itself.

**Location Scopes** describe where fixes occur, including In-Project Files (94.1%), Runtime Files (2.2%), Third-Party Files (2.1%), and User-Authored Files (3.0%). The percentages exceed 100% because an issue may span multiple scopes. For example, in panel (a) of Figure 2, an issue reported in the `pytorch/pytorch` project is ultimately traced to an implementation bug in the external dependency `facebookresearch/InterHand2.6M`, which is classified as a Third-Party File because it belongs to an external project rather than the PyTorch repository itself.

**Location Types** describe file categories affected in resolution: Code (80.8%), Test (23.5%), Configuration (15.2%), Documentation (23.5%), and Asset (4.4%). The total exceeds 100% because an issue resolution may involve multiple file types. For example, in panel (b) of Figure 2, the issue resolution location involves `gpt_engineer/preprompts/spec`, which is classified as an Asset because it is prompt text used as input rather than executable code.

## 4 COMPARISON WITH STATE-OF-ART LOCATION BENCHMARK

### 4.1 STATE-OF-ART LOCATION BENCHMARK FROM GITHUB ISSUES

**SWE-Bench Lite** Jimenez et al. is a widely adopted benchmark that collects GitHub issues together with the corresponding code patches. It contains 300 issues from 12 Python projects. However,

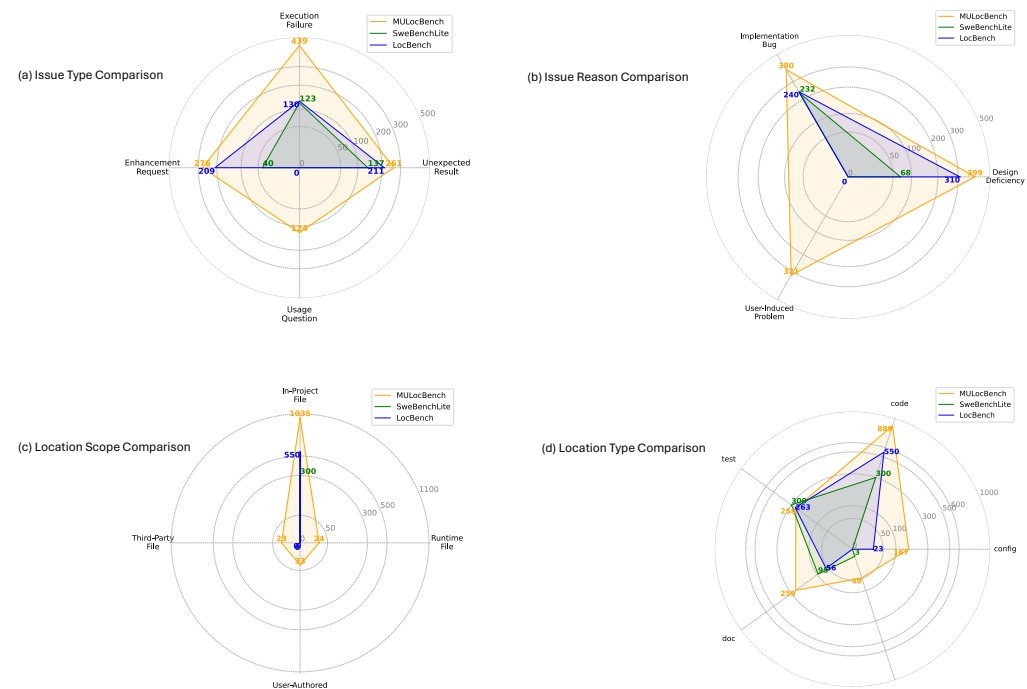

Figure 3: Issue number comparison between MULocBench, SWE-Bench Lite and LocBench

its collection only includes pull requests that modify Python code and the corresponding test files, which limits its ability to capture the broader context of real-world issue resolution.

**LocBench** Chen et al. (2025) is a new published benchmark that covers more diverse issues and a larger set of Python projects. Its latest release contains 560 issues, but we identified 10 cases where the pull requests do not correspond to the issues or are duplicated, leaving 550 valid issues from 163 repositories. Similar to SWE-Bench Lite, it only includes issues with pull requests that modify Python code, but not necessarily the associated test files.

**MULocBench**. In contrast, our MULocBench is constructed by randomly sampling issues from the top-50 most-starred Python projects, resulting in 1,100 issues from 46 Python projects. It includes resolution evidence from pull requests, commits, and comments, and covers more diverse issue types and project locations to better reflect real-world development practices.

## 4.2 LOCATION BENCHMARK COMPARISON FOR ISSUE RESOLUTION

To better understand the differences among benchmarks, we conduct a systematic comparison of MULocBench, SWE-Bench Lite, and LocBench along four dimensions: issue types, root causes, location scopes, and location types. Figure 3 shows the comparison results across these benchmarks.

**Issue Type:** SWE-Bench Lite mainly covers Execution Failures (123 issues) and Unexpected Results (137), with limited Enhancement Requests (40) and no Usage Questions (0). LocBench expands the scope by significantly increasing the number of Unexpected Results (211) and Enhancement Requests (209), while Execution Failures (130) remain similar and Usage Questions (0) are still absent. In contrast, MULocBench provides a comprehensive coverage across all four types, including Execution Failures (439), Unexpected Results (261), Enhancement Requests (276), and Usage Questions (124 issues).

**Issue Cause:** SWE-Bench Lite is dominated by Implementation Bugs (232 issues), with limited Design Deficiencies (68) and no coverage of User Problems (0), implicitly assuming that issues mainly stem from project defects. LocBench shifts the distribution, reporting 310 issues in Design Deficiencies and 240 issues in Implementation Bugs, but still excludes User Problems (0 issues), thus overemphasizing design flaws while ignoring user-side challenges. In contrast, MULocBench achieves broader balance by covering 380 issues in Implementation Bugs, 399 issues in Design Deficiencies, and 321 issues in User Problems, recognizing that many issues also arise from user misunderstandings or misuse rather than project-side faults.

**Location Scope:** SWE-Bench Lite (300 issues) and LocBench (550 issues) only include In-Project Files, ignoring all other contexts. In contrast, MULocBench not only contains a larger number of In-Project Files (1,035 issues) but also incorporates 80 issues beyond the project, including 23 in Third-Party Files, 24 in Runtime Files, and 33 in User-Authored Files. This broader coverage makes MULocBench better aligned with real-world maintenance scenarios, where issues can extend beyond the project's static code base.

**Location Type:** Both SWE-Bench Lite and LocBench are largely code-centric, with all issues involving code files. SWE-Bench Lite also includes 300 issues in test files, while its coverage of other file types is limited: 95 in documentation, 3 in assets, and none in configuration files. LocBench shows a similar pattern, with 263 issues in test files, only 56 in documentation, 23 in configuration, and none in assets. In contrast, MULocBench offers a much richer distribution of location files: 889 issues in code files, 258 in test files, 259 in documentation, 167 in configuration, and 48 in assets. This diversity better reflects the multifaceted composition of real-world projects, where issue resolution often extends beyond source code to involve configurations, documentation, and supporting resources.

Overall, both SWE-Bench Lite and LocBench remain narrow in scope: they emphasize specific issue types, focus heavily on project-side defects, restrict locations to in-project files, and are largely code-centric in artifact coverage. In contrast, MULocBench achieves broader and more balanced coverage across issue types, causes, scopes, and file categories, offering a more faithful representation of real-world issue resolution scenarios.

# 5 EFFECTIVENESS OF STATE-OF-THE-ART APPROACHES IN CODE LOCALIZATION ON MULocBench

With the popularity of SWE-Bench Lite, state-of-the-art approaches have been widely studied in the context of Python code localization. To assess their generality, we evaluate these approaches on MULocBench using in-project Python code files, since they are limited to handling such files.

## 5.1 EXPERIMENTAL SETTINGS

***Large Language Models.*** We select three leading large language models, gpt-4o-mini-2024-07-18 (GPT-4o-mini) gpt (2025), claude-3-5-haiku-20241022 (Claude3.5) cla (2025) and DeepSeekR1 Bi et al. (2024). All models support extended context windows, making them well-suited for our tasks that involve long queries and large software projects. Moreover, they are widely adopted and acknowledged by recent studies in software engineering, demonstrating strong reasoning capabilities and retrieval-augmented performance in similar scenarios Chen et al. (2025); Xia et al. (2024); Jimenez et al.; Reddy et al. (2025). We set the temperature to 0 to ensure that the outputs are mostly deterministic.

***Approaches.*** There are three main types of state-of-the-art localization approaches:

(1) Retrieval-based approach: We adopt BM25 Robertson et al. (2009), a robust and simple term-matching method widely used in software engineering tasks such as code search and issue localization Jimenez et al.; Chen et al. (2025).

(2) Procedure-based approach: Agentless Xia et al. (2024) is the most effective procedure-based issue localization approach to date. It performs hierarchical localization using a combination of prompting-based and embedding-based retrieval empowered by LLMs.

(3) Agent-based approach: We select two advanced and widely used agent-based approaches: LocAgent Chen et al. (2025) and OpenHands Wang et al. (2024). LocAgent proposes a graph-oriented LLM-agent framework for issue localization and is currently the most effective agent-based localization method. OpenHands is a React-style agent framework that enables LLMs to invoke tools for iterative reasoning and action to resolve issues.

***Metrics.*** Following prior state-of-the-art approaches Chen et al. (2025); Xia et al. (2024), which treat issue localization as a ranking task, we evaluate performance using $Top@kAcc$, where $k$ is set to 1 and 5, reflecting practical tolerance levels in real-world settings. This metric deems localization successful if all relevant code locations are correctly identified within the top-k results.

| Type | LLM | File(%) | | | | | Class(%) | | | | | Function(%) | | | | |
|---|---|---|---|---|---|---|---|---|---|---|---|---|---|---|---|---|
| | | Acc@1 | Acc@5 | P | R | F1 | Acc@1 | Acc@5 | P | R | F1 | Acc@1 | Acc@5 | P | R | F1 |
| BM25 | | 9.4 | 9.4 | 16.4 | 11.5 | 13.5 | 7.0 | 8.0 | 12.1 | 8.2 | 9.8 | 2.9 | 4.6 | 5.6 | 5.9 | 5.8 |
| Agentless | GPT-4o-mini | 10.2 | 18.6 | 15.1 | 13.1 | 14.1 | 5.8 | 9.5 | 11.8 | 7.8 | 9.4 | 4.1 | 8.7 | 5.1 | 8.0 | 6.3 |
| | Claude-3.5 | 12.2 | 23.9 | 18.6 | 15.6 | 17.0 | 3.3 | 4.5 | 12.7 | 2.5 | 4.2 | 6.4 | 11.0 | 11.2 | 5.3 | 7.2 |
| | DeepSeekR1 | 6.1 | 19.4 | 19.2 | 17.2 | 18.1 | 6.1 | 6.1 | 11.9 | 3.5 | 5.4 | 6.7 | 7.8 | 11.2 | 3.9 | 5.8 |
| LocAgent | GPT-4o-mini | 11.7 | 16.1 | 38.0 | 12.9 | 19.2 | 6.6 | 10.8 | 19.6 | 7.2 | 10.6 | 4.3 | 9.4 | 23.5 | 6.6 | 10.3 |
| | Claude-3.5 | 28.5 | 35.2 | 49.6 | 30.9 | 38.1 | 17.6 | 25.9 | 30.8 | 18.9 | 23.5 | 14.9 | 21.9 | 29.5 | 13.5 | 18.5 |
| | DeepSeekR1 | 21.0 | 29.0 | 54.7 | 39.6 | 46.0 | 27.9 | 28.2 | 28.4 | 25.8 | 27.0 | 15.2 | 25.0 | 25.0 | 16.8 | 20.1 |
| OpenHands | GPT-4o-mini | 14.0 | 24.1 | 39.2 | 16.7 | 23.5 | 11.6 | 16.1 | 22.8 | 11.8 | 15.6 | 7.7 | 12.8 | 17.1 | 8.1 | 11.0 |
| | Claude-3.5 | 24.1 | 33.5 | 49.5 | 26.2 | 34.3 | 21.3 | 28.5 | 36.2 | 20.4 | 26.1 | 11.8 | 22.0 | 23.8 | 13.8 | 17.5 |
| | DeepSeekR1 | 15.0 | 24.0 | 41.1 | 19.2 | 26.2 | 23.5 | 25.0 | 27.9 | 12.3 | 17.0 | 16.3 | 18.5 | 18.2 | 9.0 | 12.0 |

Table 1: Performance of state-of-the-art methods on Python code localization in MULocBench.

To allow a more fine-grained assessment of localization quality, we also compute precision, recall, and F1-score. For an issue, a true positive occurs when a location given by an approach exists in the benchmark. A false positive occurs when a location given by an approach does not exist in the benchmark. A false negative occurs when a configuration not given by an approach exists in the benchmark. We denote the number of true positives, false positives and false negatives as $TP$, $FP$ and $FN$. We calculate the precision as $P = \frac{TP}{TP+FP}$, the recall as $R = \frac{TP}{TP+FN}$, and the F1-score as $F1 = \frac{2 \times P \times R}{P+R}$.

Similar to previous localization tasks, we evaluate all metrics at three location levels: file, class, and function. Although function-level localization is the preferred granularity among developers Kochhar et al. (2016), as it is efficient for later debugging or program repair Yang et al. (2024a); Li et al. (2019); Liu et al. (2025). Line-level metrics also offer diagnostic value by revealing how close the model's prediction is to the true faulty statement, and the corresponding results are presented in Appendix A.7.

***DataSet.*** Since state-of-the-art approaches Chen et al. (2025); Xia et al. (2024); Reddy et al. (2025); Jimenez et al. typically localize only in-project Python files, even though pull requests may also modify other files such as JavaScript, documentation, or configuration, we follow the same setting and retain only locations in in-project Python code files, resulting in 842 issues[1].

## 5.2 RESULT

***Approach Comparison.*** Table 1 shows the performance of the state-of-the-art methods on the Python-only subset of the MULocBench at three localization levels: file, class, and function. Each method is evaluated using Acc@1, Acc@5, precision, recall, and F1-score. Performance basically degrades as the required localization granularity becomes finer (File > Class > Function). The four approaches show different performance: **LocAgent and OpenHands** yield the best and comparable results, followed by Agentless and then BM25. The best results only reach 35.2% Acc@5 / 46.0% F1 at the file level, 28.5% Acc@5 / 27.0% F1 at the class level, and 25.0% Acc@5 / 20.1% F1 at the function level. Notably, LocAgent and OpenHands with Claude3.5 achieves over 80%, 70%, 60% Acc@5 at file, class and function levels on SWE-Bench Lite Chen et al. (2025), but with limited performance on our MULocBench. This suggests that the limited generality of state-of-art approaches on issue localization. The analysis result of each approach is provided in Appendix A.5

***Model Comparison.*** **Claude 3.5 outperforms GPT-4o-mini** across all metrics in most LLM-based approaches and localization levels. An exception occurs under the Agentless at the class level, where Claude 3.5 underperforms GPT-4o-mini by 2.5% in Acc@1, 5.0% in Acc@5, and 5.2% in F1. **DeepSeek-R1** exhibits **highly variable performance** across different approaches: for example, it is the weakest model under the Agentless, while showing moderate performance in OpenHands and LocAgent. The result indicates that the limited marginal benefit of the reasoning model on state-of-art models.

***Performance Comparison across Issue Types.*** Figure 4 shows the performance comparison of different issue types on MULocBench with only Python files. **Execution Failures** achieve moderate results, with LocAgent+Claude3.5 reaching up to 41% Acc@5 and 38% F1 at the file level, and their performance still competitive 26%-32% Acc@5 and 20%-24% F1 finer levels. **Unexpected**

---

[1]Due to budget and slow inference speed of reasoning models are very slow, we randomly evaluated 285 issues for DeepSeekR1.

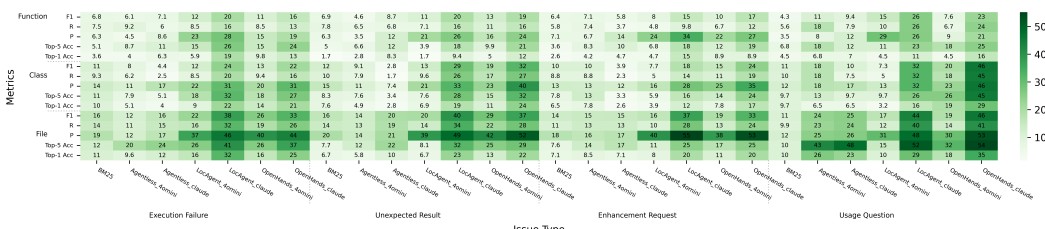

Figure 4: Performance comparison of different issue types on MULocBench with Python files.

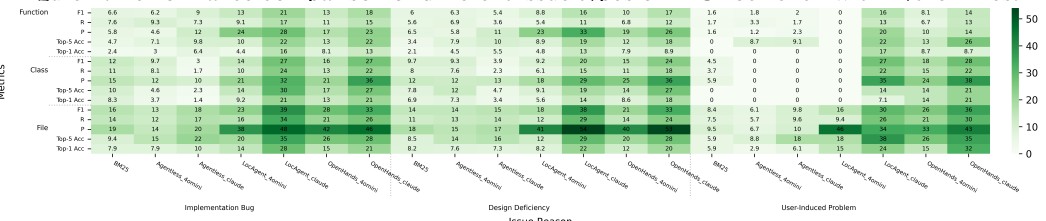

Figure 5: Performance comparison of different issue reasons on MULocBench with Python files.

**Results and Enhancement Requests** yield the weakest outcomes, with the highest Acc@5 capped at around 30%. Particularly, the precision is comparable to or even higher than that of other issue types. It indicates that while models often miss some relevant locations (low recall), the predicted locations have similar confidence to those of other issue types. **Usage Questions** achieve the strongest performance, with 54% Acc@5 and 46% F1 at the file level, likely because usage-related issues are described more concretely, providing clearer cues for localization.

*Peformance Comparison across Issue Reasons.* Figure 5 shows the performance comparison of different issue reasons on MULocBench with only Python files. **Implementation Bugs** show the most stable performance, with LocAgent+Claude 3.5 achieving the highest results: up to 35% Acc@5 and 39% F1 at the file level. Even at the class and function levels, performance remains relatively stable, with Acc@5 and F1 falling in the 22%–30%. This result reflects the fact that most existing datasets and methods in software engineering research are concentrated on bug-fixing scenarios. **Design Deficiencies** show the weakest results overall, with file-level Acc@5 around 29%. However, the precision is relatively high (29%-54%) at three levels, comparable to or even better than other issue reasons. This aligns with the nature of enhancement request issues. **User-Induced Problems** show intermediate performance, with LocAgent + Claude 3.5 achieving the best file-level result of 38% Acc@5. At the class and function levels, however, performance remains modest, with Acc@5 and F1 ranging from 14% to 27%.

## 6 EFFECTIVENESS OF LLMS IN ISSUE LOCATION ON FULL MULOCBENCH

State-of-the-art localization techniques are restricted to analyzing source code written in a single programming language (e.g., Python) and are constrained to the project repository. However, as reflected in the MULocBench benchmark, real-world issue localization is far more complex: relevant locations may appear in non-code files, span multiple file types, or even lie outside the scope of the project. Motivated by the success of LLMs, we investigate their effectiveness on the full MULocBench, which reflects realistic issue localization scenarios.

### 6.1 EXPERIMENTAL SETTINGS

The evaluation metrics and LLMs used here are the same as in Section 5.

*LLM-based Approaches* To systematically explore the capabilities of LLMs in localization, we design five LLM-based approaches that represent different levels of information availability and reasoning complexity. These variants are inspired by common strategies in software engineering and recent successes of LLMs in code-related tasks . The prompts are shown in Appendix A.12

**(1) Closed-Book LLM:** In the closed-book setting, we directly prompt LLMs to generate localization predictions for GitHub issues by only providing Github issue report. This approach evaluates the model's ability to rely solely on its prior knowledge.

| Type | LLM | File(%) | | | | | Class(%) | | | | | Function(%) | | | | |
|---|---|---|---|---|---|---|---|---|---|---|---|---|---|---|---|---|
| | | Acc@1 | Acc@5 | P | R | F1 | Acc@1 | Acc@5 | P | R | F1 | Acc@1 | Acc@5 | P | R | F1 |
| ClosedBook | GPT-4o-mini | 10.4 | 15.1 | 15.8 | 10.0 | 12.3 | 6.8 | 7.9 | 9.0 | 4.4 | 5.9 | 6.1 | 8.6 | 5.8 | 2.3 | 3.3 |
| | Claude3.5 | 14.5 | 20.0 | 23.7 | 15.2 | 18.5 | 8.9 | 9.5 | 17.7 | 4.3 | 6.9 | 8.6 | 10.2 | 8.9 | 2.8 | 4.3 |
| | DeepSeekR1 | 15.0 | 25.0 | 24.2 | 20.1 | 22.0 | 14.3 | 16.3 | 11.8 | 10.2 | 11.0 | 6.5 | 16.1 | 8.3 | 6.1 | 7.0 |
| ProStructure | GPT-4o-mini | 12.4 | 17.9 | 17.5 | 12.9 | 14.9 | 7.9 | 9.1 | 8.3 | 6.9 | 7.6 | 6.2 | 9.1 | 4.9 | 3.9 | 4.3 |
| | Claude3.5 | 17.4 | 27.0 | 27.4 | 18.6 | 22.1 | 9.3 | 10.5 | 23.9 | 8.3 | 12.4 | 8.6 | 11.0 | 11.0 | 3.9 | 5.8 |
| | DeepSeekR1 | 13.0 | 29.0 | 19.5 | 18.2 | 18.8 | 16.3 | 24.5 | 13.6 | 11.4 | 12.4 | 3.2 | 12.9 | 6.8 | 4.8 | 5.7 |
| Hint | GPT-4o-mini | 13.5 | 22.3 | 16.6 | 15.6 | 16.1 | 6.8 | 10.1 | 7.4 | 6.8 | 6.8 | 6.5 | 9.9 | 3.8 | 3.6 | 3.7 |
| | Claude3.5 | 18.6 | 32.5 | 30.6 | 20.2 | 24.4 | 11.2 | 14.1 | 17.3 | 8.6 | 11.5 | 9.8 | 13.1 | 9.0 | 5.1 | 6.5 |
| | DeepSeekR1 | 18.0 | 35.0 | 23.8 | 23.0 | 23.4 | 16.3 | 24.5 | 12.5 | 10.8 | 11.6 | 16.1 | 21.0 | 10.2 | 7.1 | 8.3 |
| Pipeline-ProStructure | GPT-4o-mini | 10.4 | 14.9 | 14.5 | 9.9 | 11.8 | 6.6 | 7.6 | 11.9 | 5.2 | 7.3 | 3.2 | 4.8 | 6.3 | 3.0 | 4.1 |
| | Claude3.5 | 16.9 | 23.3 | 27.2 | 16.9 | 20.9 | 10.9 | 12.0 | 19.3 | 8.7 | 12.0 | 8.1 | 9.4 | 12.6 | 5.9 | 8.0 |
| | DeepSeekR1 | 17.0 | 33.7 | 24.2 | 19.0 | 21.2 | 12.2 | 24.5 | 12.1 | 8.4 | 9.9 | 12.9 | 19.6 | 10.2 | 5.5 | 7.1 |
| Pipeline-Hint | GPT-4o-mini | 10.7 | 17.3 | 16.7 | 11.7 | 13.8 | 6.8 | 8.5 | 13.4 | 5.9 | 8.2 | 4.2 | 6.1 | 7.8 | 3.4 | 4.8 |
| | Claude3.5 | 17.7 | 24.9 | 28.6 | 18.5 | 22.5 | 12.6 | 14.5 | 26.2 | 10.2 | 14.7 | 8.6 | 11.2 | 14.5 | 6.5 | 9.0 |
| | DeepSeekR1 | 13.0 | 32.0 | 25.9 | 19.0 | 21.9 | 12.2 | 20.4 | 14.0 | 9.6 | 11.4 | 8.1 | 12.9 | 9.8 | 5.5 | 7.0 |

Table 2: Performance comparison with LLM-based approaches on full MULocBench

**(2) Project-Structure LLM:** Directly feeding the entire project content into the LLM is computationally infeasible due to input length limitations. Given that our task is localization, we instead provide the model with project structure information to assist LLM' reasoning. The project structure includes a hierarchical representation of the directory tree, along with file names, file extensions, and their relative paths.

**(3) Location-Hint LLM:** Software projects often contain hundreds or thousands of files, which makes localization difficult due to the large and noisy search space. We give the LLM location scope (e.g., within-project, third-party, user-authored files) and location type (e.g., code, configuration, documentation) as hints. We also provide the project structure but filter out files that do not match the specified location type, so the model can search within a smaller and more relevant subset.

**(4) Pipeline (Project-Structure Guided) LLM:** A common way to apply LLM to complex tasks is to split the process into sequential steps. Following this idea, we design a two-stage pipeline for issue localization. In the first stage, we provide the issue report together with the project structure to let the LLM predict the candidate files. In the second stage, we give the predicted files along with their class and function names to the LLM to identify finer-grained elements such as the specific class, functions, and line numbers. We deliberately avoid supplying full file content since it often exceeds context length, slows inference, and yields limited additional benefit.

**(5) Pipeline (Location-Hint Guided) LLM:** Similar to the structure-guided pipeline, this approach decomposes issue localization into two stages. In the first stage, we provide the issue report together with the location scope and location type as hints, and restrict the project structure to only include files matching the given location type. The LLM then predicts files based on the prompt. The second stage follows the same procedure as the second stage of the pipeline structure–guided LLM.

## 6.2 RESULT

*Approach Comparison.* Table 2 shows the performance of the LLM-based methods on the full MULocBench at three localization levels: file, class, and function. At the file level, **Location-Hint** achieves the best results, with 35.0% Acc@5 and 24.4% F1. At the class and function levels, **Location-Hint** yields the highest Acc@5 (24.5% and 21.0%, respectively), while **Pipeline-Location-Hint** achieves the best F1 (14.7% and 9.0%). The superior F1 of the pipeline variant can be attributed to its notably higher precision. By narrowing down the candidate search space to fewer, more relevant files, the pipeline approach reduces noise and minimizes false positives, which directly improves precision and consequently boosts the F1 score.

**Performance improves steadily from Closed-Book to Project-Structure to Location-Hint**, demonstrating that richer structural and contextual information facilitates more accurate localization. In contrast, **pipeline strategies** exhibit trade-offs: they decrease file-level accuracy but can improve precision at the class and function levels by limiting predictions to earlier-selected files. This suggests that pipeline strategies are not uniformly effective, as accurate localization requires integrating information across levels rather than following rigidly staged processes.

*Model Comparison.* Across all approaches and granularities, DeepSeek-R1 and Claude 3.5 consistently outperform GPT-4o-mini. DeepSeek-R1 surpasses Claude 3.5 on the Acc@k metric, but Claude 3.5 achieves higher F1 scores. This indicates that DeepSeek-R1 is better at coarse-grained

candidate ranking, whereas Claude 3.5 provides more precise and balanced predictions, leading to stronger overall localization quality.

***Performance Comparison across Issue Types and Reasons.*** Figure 7 and Figure 8 shows the performance comparison of different issue types and reasons on the full MULocBench. The performance differences observed on Python code closely mirror those on the full benchmark. The details can seen in Appendix A.9.

***Performance Comparison across Location Scopes.*** Figure 9 presents the performance comparison across different location scopes on the full MULocBench. **In-Project Files** show the strongest performance, with up to 34% Acc@5 and 22% F1 at the file level, and non-negligible results at finer granularities (14% Acc@5 and 13% F1 at the class level and 13% Acc@5 and 8.5% F1 at the function level). **User-Authored Files** reach up to 25% Acc@5 at the file level, while all other metrics remain zero. **Runtime Files** yield limited performance, with the best file-level results around 10% Acc@5 and F1. **Third-Party Files** perform worst, with all metrics at zero, as external dependencies are rarely referenced in issue descriptions and lack sufficient project-specific context. These findings indicate that LLM-based localization is effective on project-maintained code but struggles with external or indirect files.

***Performance Comparison across Location Types.*** Figure 10 presents the performance comparison across different location types on the full MULocBench. Overall, Code and Configuration Files achieve stronger performance than the other types: for **Code Files**, ProjectStructure reaches up to 41% Acc@5 and 30% F1 at the file level, and Location-Hint achieves 18%–26% Acc@5 at class and function levels. Notably, Location-Hint performs slightly lower than ProjectStructure at the file level, with a gap of about 2%. This difference is reasonable because Location-Hint distributes its strength more evenly across various file types (e.g., configurations and tests), rather than being specialized for source code as ProjectStructure is. For **Configuration Files**, Location-Hint achieves the best results with 30% Acc@5 and 26% F1 at the file level. **Test Files** show moderate performance, with Location-Hint achieving 29% Acc@5 and 22% F1 at the file level, but class- and function-level F1 often falls below 5% due to very low recall. **Documentation Files** yield limited performance, with the best results around 22% Acc@5 and 20% F1. **Asset Files** show similarly limited result, with 20% Acc@5 and 21% F1. These results indicate that LLM-based localization is more effective for code-related files, whereas performance drops for non-code files.

# 7    FAILURE ANALYSIS

To investigate why LLMs fail to perform accurate issue localization, we select five issues for which state-of-the-art methods or LLMs predict incorrect locations. The full examples are provided in Appendix 7. The failure analysis for these cases is detailed as follows:

- **Word-Matched Misleading**: Rather than performing true causal reasoning, LLMs tend to rely on keywords in the issue description to identify locations such as files, classes, or functions. This keyword matching can be misleading, often resulting in incorrect localization. For example, in Figure 11, an issue states that "module 'torch.cuda' has no attribute 'comm' keywords". LLMs suggest a list of files containing the keywords such as "torch" and "cuda". However, the correct file is `common/nets/module.py`, which diverges from these keyword-matched files.

- **Code-Centric Bias**: Influenced by the predominance of code in software projects, LLMs tend to assume that issues correspond to code-level problems. This assumption drives reasoning in the wrong direction and produces mislocalization. For example, in Figure 12, an issue states "load model failed", LLMs respond by suggesting code-related locations, whereas the correct locations are configuration-related files.

- **Responsibility Chain Tracking Limitation** Issue Localization follows a small, sequential responsibility path, but LLMs frequently fail to trace this multi-step chains. Instead of following the causal flow to downstream modules, they over-predict files that are heuristically associated with the issue or frequently co-occur in the project. For example, in Figure 13, the Flask issue "Switch to importlib breaks scripts with app.run()", LLMs correctly identify `src/flask/scaffold.py` but miss the downstream critical file `src/flask/helpers.py`. At the same time, they over-predict files such as `app.py`, due to the fact that `app.py` often interacts with core modules like `scaffold.py`.

Based on the three identified limitations of LLMs, we propose three promising directions for future work on improving issue localization:

- **Unrelated keywords masking.** Masking words reduces spurious lexical overlaps between issue descriptions and project locations(e.g., file names) so that LLMs can rely more on semantic cues rather than shallow word matching.
- **File-type–aware keyword mapping.** Construct mapping libraries that associate each file type with representative keywords.These mappings can be used as a part of prompt to alleviate code-centric bias and guide models toward non-code or mixed-context issues.
- **Project-level responsibility chains construction** Construct project-level responsibility chains that explicitly encode relationships among entities (project/file/class/function) to guide LLMs to perform deterministic deep reasoning.

## 8 RELATED WORK

**Project Benchmarks for Evaluating LLMs.** With the rise of large language models (LLMs), several benchmarks have emerged to evaluate their performance Jimenez et al.; Chen et al. (2025); Zhuo et al. (2024); Deng et al. (2025); Niu et al. (2023); Chen et al. (2021); Ouyang et al. (2024); Gao et al. (2023); Jiang et al. (2024); Jain et al. (2024); Mündler et al. (2024); Xie et al. (2024). SWE-Bench Jimenez et al., collects 2,294 issues with pull requests from 12 Python projects, mainly focusing on bug fixing. SWE-Bench Lite provides a subset of 300 issues to encourage adoption, which is widely used as a standard benchmark for research. Recently, LocBench Chen et al. (2025) expands coverage to 560 issues with more diverse issue types. These benchmarks primarily focus on Python code and rely heavily on pull requests as the sole resolution signal. In reality, software project issues can be addressed through pull requests, commits, or comments. To better reflect these real-world scenarios, our MULocBench collects 1,100 issues accompanied by pull requests, commits, or comments, offering broader and more balanced coverage across issue types, causes, location scopes, and types.

**Issue Localization.** Issue localization refers to the set of code or non-code artifacts (e.g., source files, test files, configuration files, documentation) that are relevant for resolving software issues. Traditional retrieval-based methods Robertson et al. (1994); Wang et al. (2022); Suresh et al. (2024), such as BM25, rely on lexical matching to return a ranked list of potentially relevant files. Recently, LLM-based retrieval methods have been proposed to improve localization performance Xia et al. (2024); Chen et al. (2025); Reddy et al. (2025); Wang et al. (2024); Yang et al. (2024b); Jiang et al. (2025); Zhang et al. (2024a); Tao et al. (2024); Xie et al.; Ma et al. (2025). For example, Agentless Xia et al. (2024) adopts a three-phase approach to identify relevant code locations. LocAgent Chen et al. (2025) further introduces a heterogeneous graph representation to enhance code localization. Currently, LocAgent achieves over 70% top-1 accuracy on file-, class-, and function-level localization on SWE-Bench Lite. However, when evaluated on MULocBench, LocAgent achieves less than 40% Acc@5. This highlights the increased difficulty and broader diversity of our benchmark, as well as the limitations of existing methods in generalizing to more realistic and complex issue scenarios.

## 9 CONCLUSION

In this paper, we present MULocBench, a comprehensive project location benchmark for issue resolution. Compared with existing benchmarks, MULocBench covers more diverse issue types, root causes, location scopes, and file types, providing a richer basis for evaluation. Our experiments with state-of-the-art and LLM-based approaches reveal that even the best methods achieve below 40% Acc@5 at the file level, highlighting substantial challenges and the need for further advances in realistic issue resolution.

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

# A APPENDIX

## A.1 ISSUE FILTERING PROCESS

**Issue filtering:** Reliable location information is critical for our study, but many issues lack such information due to being open, unresolved, or having ambiguous descriptions. To obtain reliable location data, we apply three filtering conditions: selecting closed issues, retaining those with confirmed resolutions, and keeping only issues with clear location information. After filtering, there are 1,100 issues from 46 projects. Among them, 716 have pull requests, 63 have commits, and 321 have resolution-related comments. The details are as follows:

- *Select closed issues:* We exclusively consider closed issues, as open issues typically mean ongoing discussions or incomplete resolutions.

- *Determine resolution status:* For closed issues, we further identify whether they are resolved. We observe three common ways to determine resolution status:

  - The issue has a corresponding merged pull request (PR).
  - The issue is referenced or closed by a commit that has been integrated into the main branch.
  - The issue comments include confirmation, such as "thank you, it answered my question," indicating the issue has been resolved.

- *Check location clarity:* We perform this check only for issues without linked pull requests or commits, as those contain location information. Four annotators independently record clear locations in the resolution comments, and any discrepancies from resolution comments or location are resolved through discussion. A location in resolved comments is labeled as clear if it is searchable in the project. Otherwise, it is labeled as unclear. For example, for an issue, whose resolution comment is "got it working, just changed ... in `ui.py`", and the `ui.py` is in the project, so we think the location is clear.

## A.2 MULocBench Statistics

**Project and Issue Statistics**. Figure 6 shows the issue counts of the 46 projects included in MULocBench. The dataset covers a wide range of project types and domains. For example, pandas and scikit-learn represent popular data processing and machine learning libraries, transformers and keras correspond to deep learning frameworks, while ansible and yt-dlp reflect automation and media processing tools. Smaller projects such as DeepSpeeck-V3 and grok-1 illustrate that MULocBench also includes niche and emerging repositories. This diversity highlights the benchmark's coverage of different programming scenarios, user communities, and issue characteristics.

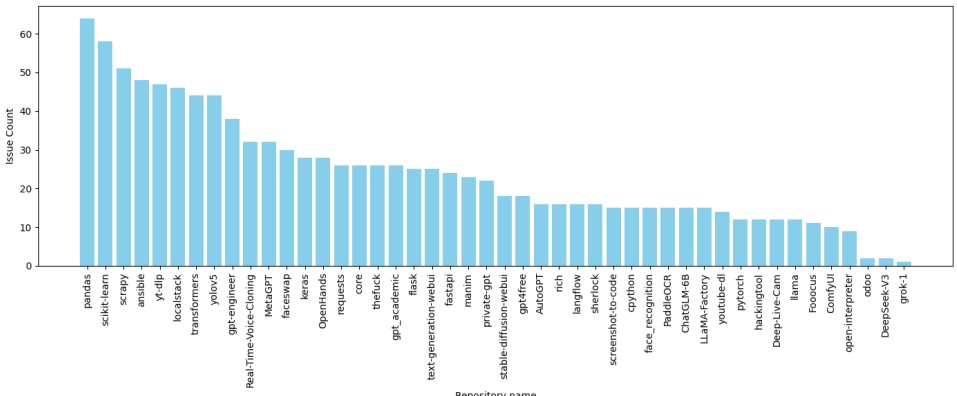

Figure 6: Statistics of issue counts across each projects.

| File Type | Issue Number | File Number |
|---|---|---|
| Python | 842 | 2494 |
| OtherPL (JavaScript/C/Java) | 16 | 53 |
| Test | 273 | 513 |
| Documentation | 259 | 443 |
| Asset | 48 | 59 |
| Configuration | 167 | 308 |

Table 3: Statistics of file types in the MULocBench.

**File Type Distribution in MULocBench**. Table 3 summarizes the distribution of file types in MULocBench. Among the 1,100 issues, although the projects are primarily Python-based, not all issue localizations involve Python code. This distribution indicates that issue localization tasks are not limited to Python source code but also span a diverse set of file types that play important roles in modern software projects.

Approximately 23.5% of issues correspond to locations that do not include any Python files. Specifically, 1.5% of issues involve files written in other programming languages (e.g., JavaScript, C, or Java), 24.8% involve test files, 23.5% involve documentation files, 4.4% involve asset files, and 15.2% involve configuration files.

From the file perspective, Python source files account for over 2,000 files. Each of the test, documentation, and configuration categories contains approximately 300–500 files. Files in other programming languages and asset files are less common, with each category containing slightly over 50 files.

## A.3 Correlation Analysis between Issue Characteristics and Issue Localization

To understand how issue characteristics influence localization performance, we consider nine issue characteristics to analyze their relationship with localization performance Tao et al. (2024); Kuramoto et al. (2024); Panichella et al. (2021). Table 4 presents the descriptive statistics of the issue characteristics, including their mean, variance, minimum, and maximum values. The values of is-

Table 4: Descriptive Statistics of Issue Features

| Feature | Mean | Variance | Min | Max |
|---|---|---|---|---|
| iss_descr_length | 615.45 | 2,692,121.36 | 1 | 37,442 |
| has_traceback | 0.21 | 0.17 | 0 | 1 |
| has_code_block | 0.50 | 0.25 | 0 | 1 |
| has_inline_code | 0.64 | 0.23 | 0 | 1 |
| file_num | 2.85 | 16.93 | 1 | 30 |
| cls_num | 2.11 | 35.48 | 0 | 68 |
| func_num | 2.80 | 56.46 | 0 | 114 |
| line_num | 22.09 | 8,511.70 | 0 | 1,291 |
| diff_file_type_num | 1.49 | 0.58 | 1 | 5 |

sue characteristics exhibit high variability, reflecting the diverse nature of issues and their potential impact on localization difficulty.

- **iss_des_length**: The number of words in the issue description. Longer descriptions generally provide more detailed information for developers to understand the problem context, interpret the symptoms, and identify potential root causes.

- **has_traceback**: Indicates whether the issue description contains an execution traceback. Tracebacks provide precise failure points, including file paths, line numbers, and call chains, which usually allow developers more quickly identify the root cause.

- **has_code_block**: Indicates whether the issue description contains a Markdown code block ("'). Code blocks often include reproduction scripts, configuration snippets, error logs, or minimal examples. Such explicit technical content provides structured clues about the execution context, helping narrow down the related components or functions in the system.

- **has_inline_code**: Indicates whether the issue description contains inline code segments (enclosed in backticks, e.g., function name). Inline code is typically used to reference specific identifiers, such as variables, function names, file paths, or configuration keys. These references add specificity to the description, improving the alignment between the issue text and concrete code elements involved in localization.

- **file_num**: The number of files involved in issue localization. More files indicate a wider scope and greater structural complexity, making localization more challenging.

- **cls_num**: The number of classes involved in issue localization. More classes suggest cross-cutting changes and higher design complexity, increasing the difficulty of understanding and localizing the issue.

- **func_num**: The number of functions involved in issue localization. A larger number of functions implies multiple logical units are affected, making it harder to identify the root cause.

- **line_num**: The total number of lines changed involved in issue localization. Larger changes reflect greater issues complexity and increase the search space for locating the source of the issue.

- **diff_file_type_num**: The number of different file types modified (e.g., source code, tests, documentation). A higher number indicates that cross-cutting changes require diverse expertise, making the issue harder to locate.

We assess the relationship between the issue characteristics and the issue localization by cal- culating their correlation coefficient. Given that the distribution of these characteristics exhibits skew- ness, and the localization result is binary (correct or not), logistic regression is employed for the analysis across three LLMs. Table 5 and Table 6 shows the correlation between issue characteristics and issue localization in SOTA approaches and LLM-prompting approaches.

Location-related features, especially the number of files and different file types, consistently show strong negative correlations across all LLMs, indicating that **issues involving more extensive locations are generally harder to localize**.

| Approach | | iss_des_length | has_traceback | has_code_block | has_inline_code | file_num | cls_num | func_num | line_num | diff_file_num |
|---|---|---|---|---|---|---|---|---|---|---|
| BM25 | | 0.0001* | -0.0423* | 1.1124* | 1.5827* | -0.4134* | -0.1683* | -0.239* | -0.0014* | -0.884* |
| Agentless | GPT-4o-mini | 0.0001* | -0.1558* | -0.1712* | -0.0277* | -0.4276* | -0.1159* | -0.1033* | -0.0028* | -0.9445* |
| | Claude3.5 | 0.0002* | -0.008* | 0.1011* | 0.2063* | -0.803* | -0.2868* | -0.2503* | -0.0014* | -1.1176* |
| | DeepSeekR1 | -0.0054* | -0.9902* | -0.8519* | -0.3853* | -1.0402* | -0.7888* | -1.2239* | -0.0638* | -1.5802* |
| LocAgent | GPT-4o-mini | 0.0001* | 0.8571* | 1.226* | 2.2032* | 0.0077* | -0.2445* | -0.2074* | -0.0135* | -0.9089* |
| | Claude3.5 | -0.0002* | -0.0128* | 0.5467* | 0.2628* | -0.8365* | -0.2712* | -0.1331* | -0.0315* | -1.9413* |
| | DeepSeekR1 | 0.0011* | 0.9508* | -0.0359* | 0.1221* | -1.602* | -0.4581* | -1.0821* | -0.1519* | -1.701* |
| OpenHands | GPT-4o-mini | 0.0001* | 0.2253* | 0.2338* | 0.4942* | -0.6249* | -0.2258* | -0.2517* | -0.0225* | -0.8046* |
| | Claude3.5 | -0.0002* | -0.0009* | -0.0752* | 0.0771* | -1.0655* | -0.4096* | -0.3868* | -0.0142* | -1.6713* |
| | DeepSeekR1 | 0.0005* | 0.5155* | -0.4818* | 0.1382* | 0.4099* | -0.6342* | -0.7769* | -0.0188* | -0.7501* |

Table 5: Correlation between issue complexity and issue localization in SOTA approaches(P-values < 0.05 are marked with *).

| Approach | | iss_des_length | has_traceback | has_code_block | has_inline_code | file_num | cls_num | func_num | line_num | diff_file_num |
|---|---|---|---|---|---|---|---|---|---|---|
| ClosedBook | GPT-4o-mini | 0.0001 | 0.3269 | 0.2364 | 0.2182 | -2.1504* | -0.2016* | -0.2477* | -0.0244* | -4.4434* |
| | Claude3.5 | 0.0003* | 0.3471* | 0.3561* | 0.3028* | -1.9109* | -0.269* | -0.2916* | -0.0316* | -3.6438* |
| | DeepSeekR1 | 0.0015* | 0.5584* | 0.9079* | 1.1474* | -1.2978* | -0.0587* | -0.1169* | -0.0094* | -1.431* |
| ProStructure | GPT-4o-mini | 0.0002* | 0.2757* | -0.1109* | 0.003* | -1.7122* | -0.1987* | -0.2417* | -0.0208* | -3.537* |
| | Claude3.5 | 0.0002* | 0.2401* | 0.0478* | 0.0671* | -2.0467* | -0.3434* | -0.3431* | -0.0332* | -3.2742* |
| | DeepSeekR1 | 0.0001* | 0.7042* | 0.6216* | 0.7548* | -2.0241* | -0.0767* | -0.1246* | -0.0154* | -2.315* |
| Hint | GPT-4o-mini | 0.0* | 0.2775* | -0.2031* | -0.1733* | -1.8291* | -0.3247* | -0.297* | -0.0231* | -2.9825* |
| | Claude3.5 | 0.0001* | 0.2551* | 0.1672* | 0.0854* | -1.3183* | -0.2462* | -0.2148* | -0.0219* | -1.7195* |
| | DeepSeekR1 | 0.0001* | 0.3434* | 0.4042* | 0.5596* | -1.2639* | -0.0724* | -0.1256* | -0.0147* | -1.284* |
| Pipeline-ProStructure | GPT-4o-mini | 0.0001* | 0.2478* | -0.1216* | -0.0484* | -1.7954* | -0.1189* | -0.1578* | -0.0204* | -2.9983* |
| | Claude3.5 | 0.0002* | 0.1686* | 0.1394* | 0.187* | -2.1249* | -0.2483* | -0.2624* | -0.0247* | -3.5681* |
| | DeepSeekR1 | 0.0002* | -0.0841* | -0.1206* | 0.2642* | -1.4023* | -0.1211* | -0.1736* | -0.0125* | -1.5113* |
| Pipeline-Hint | GPT-4o-mini | 0.0001* | 0.2315* | -0.2248* | -0.0471* | -1.7547* | -0.224* | -0.2239* | -0.0188* | -2.9143* |
| | Claude3.5 | 0.0003* | 0.1821* | 0.2549* | 0.286* | -2.1216* | -0.2834* | -0.2907* | -0.026* | -4.3645* |
| | DeepSeekR1 | 0.0002* | -0.0841* | -0.1206* | 0.2642* | -1.4023* | -0.1211* | -0.1736* | -0.0125* | -1.5113* |

Table 6: Correlation between issue complexity and issue localization in LLM-prompting approaches (P-values < 0.05 are marked with *).

Textual features show weaker and more variable correlations. Issue description length has only a weak correlation (<0.01), while richer textual information (e.g., tracebacks) provides modest correlation overall. However, the effect of richer textual features varies across LLMs. For example, Pipeline-Hint applied on GPT-4o-mini and Claude 3.5 benefits from tracebacks, whereas DeepSeekR1 is disrupted by tracebacks, resulting in negative correlations. These variable patterns show that, **although richer textual information can aid localization, current LLMs cannot fully leverage these signals across complex issue contexts, limiting their effectiveness**.

### A.4 EMPIRICAL ANALYSIS OF ISSUES AND LOCATIONS

#### A.4.1 ANALYSIS METHOD

In Section 2, we collect 1,100 issues associated with pull requests, commits, or resolution-related comments. To define taxonomy for issues or locations, we employ the card sorting approach with two iterations Zhang et al. (2024b); Zheng et al. (2025); Bi et al. (2022); Liu et al. (2024). In the first iteration, we randomly sample issues for each type with a confidence level of 95% and an error margin 5%. Then, two authors first independently analyze the issue report and then they annotate the issue with a short description. They discuss and resolve all disagreements if their descriptions do not have the same meaning. The Cohen's kappa agreement between two labels all above 0.7 (substantial agreement). Next, they work together to group all annotations into a category with the corresponding definition.

In the second iteration, two authors independently annotate the remaining issues with the predefined categories. If the remaining issues do not fit existing categories, they are annotated with new short descriptions. It is found that no new categories are needed. The Cohen's kappa agreement for issue and location category between two labels all above 0.75. Then, two authors discuss the disagreements to reach an agreement. Figure 2 shows examples of issues with different types, root causes, location scopes, and location types.

#### A.4.2 RESULT

**Issue Type** captures the kinds of problem developers raise in issue reports, such as errors during code execution. It reflects what challenges developers encounter or what support they seek during project use and maintenance. We categorize issues into four types.

(1) **Execution Failure**: Issues where the project fails to run properly, such as installation errors, startup crashes, or runtime errors. This category accounts for 39.9% (439 issues). For example, in panel (a) of Figure 2, a developer reports an AttributeError: module 'torch.cuda' has no attribute 'comm' when executing code with the `pytorch` project.

(2) **Unexpected Result**: Issues where the project runs successfully, but its behavior or output deviates from expectations, such as producing incorrect results. It accounts for 23.7% (261 issues). For example, in panel (c) of Figure 2, a developer attempts to run `Fooocus` to load the model, but the program simply exits without loading it or showing any errors.

(3) **Enhancement Request**: Issues where users request new functionality, improvements to existing features, or extensions to the project's capabilities. It accounts for 25.1% (276 issues). For example, in panel (b) of Figure 2, a developer requests remove unnecessary configurations, such as tdd.

(4) **Usage Question**: Issues where users ask about project usage, such as how to accomplish specific tasks, locating files, or understanding code structure. It accounts for 11.3% (124 issues). For example, in panel (d) of Figure 2, a developer asks how to use `Sponsorblock` as part of Python script.

**Root Causes of Issues** capture the underlying reasons behind issue reports, such as code bugs. We categorize issue causes into three types:

(1) **Implementation Bug**: Problems arising due to errors in the project's implementation, where the functionality does not perform as expected, such as logic errors and incorrect outputs. It accounts for 34.5% (380 issues). For example, in panel (a) of Figure 2, a developer fixes an AttributeError raised by calling a non-existent API `torch.cuda.comm.broadcast`.

(2) **Design Deficiency**: Situations where a project is incomplete or poorly implemented, such as missing functionality and need for refactoring. It accounts for 36.3% (399 issues). For example, in panel (b) of Figure 2, a developer removes unnecessary configurations to improve design clarity and maintainability.

(3) **User-Induced Problem**: Problems arising from incorrect usage, misunderstandings, or limited knowledge of the project on the user's side. It accounts for 29.2% (321 issues). For example, in panel (c) of Figure 2, developers fail to load a model in `Fooocus` until they correct their own configuration settings.

**Location Scope for Issue Resolution** refers to the range of files involved in issue resolution, which can also include locations beyond the project repository. To use a software project, developers depends not only on files within the project itself, but also on external dependencies, runtime-generated files, and user-authored code. We classify location scopes into four categories:

(1) **In-Project File**: Files that are physically present within the project repository. Typical examples include source code files (`main.py`), configuration files (`settings.yaml`), or documentation (`README.md`). It accounts for 94.1% (1035 issues). For example, in panel (b) of Figure 2, an issue from the `AntonOsika/gpt-engineer` project was resolved by modifying three files within the repository, which shows a case of in-project file localization.

(2) **Runtime File**: Files that are outside the project but are generated during execution or required at runtime. It accounts for 2.2% (24 issues). For example, a generated file created after running a source script, or an environment-specific configuration file such as `.env`. In panel (c) of Figure 2, an issue from the `lllyasviel/Fooocus` project falls into this category: its resolution required modifying two files, `config.txt` and `config_modification_tutorial.txt`, which are not part of the repository but are created after execution.

(3) **Third-Party File**: Files from external projects, such as third-party libraries that the current project depends on. It accounts for 2.1% (23 issues). For example, in panel (a) of Figure 2, an issue reported in the `pytorch/pytorch` project is ultimately traced to an implementation bug in the external dependency `facebookresearch/InterHand2.6M`, rather than in PyTorch itself.

**(4) User-Authored File**: Files written by users outside the original project, typically referenced in issue descriptions when reporting problems. It accounts for 3.0% (33 issues). For example, in issue 78759 of the `ansible/ansible` project, a user reports an error when passing a variable to a loop and provides the corresponding configuration. Another user suggests to replace `loop:{{sysctl.values}}` with `loop:{{sysctl['values']}}`, which resolves the issue. In this case, the location of the issue resolution is the user's own configuration file, specifically the line `loop:{{sysctl.values}}`.

**Location Type for Issue Resolution** refers to the file types affected during issue resolution. A software project is not solely code; it also depends on configurations, documentation, and other artifacts that together to function as a complete project. We categorize issue locations into five types based on their functional roles.

**(1) Code**: Source files that implement the core logic and functionality of the project, typically written in programming languages such as `.py` or `.cpp`. It accounts for 80.8% (889 issues). For example, in panel (a) of Figure 2, the resolution was located in `common/nets/module.py`, which is a Python source file.

**(2) Test**: Test files that verify the correctness of the project through test cases. This category includes unit, integration, and system test files, e.g., `test_*.py` and `*.spec.js`. It accounts for 23.5% (258 issues). For example, in panel (b) of Figure 2, the issue resolution location involves the file `tests/test_collect.py`, which is a Python test file.

**(3) Configuration**: Configuration files used for build processes, dependency management, or runtime settings, such as `requirements.txt` and `pom.xml`. It accounts for 15.2% (167 issues). For example, in panel (c) of Figure 2, the issue resolution location contains the file `presets/default.json`, which is a configuration file.

**(4) Document**: Documentation files intended to describe and support the usage and maintenance of the project, typically found in files such as `README` or within the `docs` directory. It accounts for 23.5% (259 issues). For example, in panel (d) of Figure 2, the issue resolution location involves `README.md`, which provides guidance for developers on using the project.

**(5) Asset**: Non-code resources that support system operation or enhance functionality, such as data files (e.g., `.csv`, `.json`), static resources (e.g., images, fonts), or auxiliary scripts. It accounts for 4.4% (48 issues). For example, in panel (b) of Figure 2, the issue resolution location involves `gpt_engineer/preprompts/spec`. As this file contains only prompt text consumed as input, rather than executable code, it is classified as an asset.

A.5  STATE-OF-ART APPROACHES COMPARISON

From Table 1, the four approaches show different performance: LocAgent and OpenHands yield the best and comparable results, followed by Agentless and then BM25. The details are as follows:

- **BM25:** Yields the weakest results across all levels. Acc@5 remains below 10%, and F1 stays under 14% for all levels. This confirms that lexical retrieval alone struggles with reasoning and semantic understanding.

- **Agentless:** Provides moderate improvements over BM25, with the best file-level results of 23.9% Acc@5 and 18.1% F1. However, class-level performance remains weak, with gains over BM25 limited to less than 2% Acc@5 and in some cases even lower (e.g., 4.5% Acc@5), indicating instability across granularities.

- **LocAgent:** Offers the best overall performance. At the file, class, and function levels, the best Acc@5 reaches 35.2%, 28.2%, 25.0%, and the best F1 reaches 46.0%, 27.0%, 20.1%, respectively. The result suggests that incorporating graph-based state representations helps enhance reasoning and improves localization across granularities.

- **OpenHands:** Reaches performance close to LocAgent. At the file, class, and function levels, the best Acc@5 reaches 33.5%, 28.5%, 22.0%, and the best F1 reaches 34.3%, 26.1%, 17.15%, respectively. The result indicates that structured multi step tool use can substantially enhance localization quality.

A.6   PERFORMANCE COMPARISON OF LLM-BASED APPROACHES ON FULL MULOCBENCH

Table 2 shows the performance of the LLM-based methods on the full MULocBench at three local-ization levels: file, class, and function. At the file level, **Location-Hint** achieves the best results, with 35.0% Acc@5 and 24.4% F1. At the class and function levels, **Location-Hint** yields the high-est Acc@5 (24.5% and 21.0%, respectively), while **Pipeline-Location-Hint** achieves the best F1 (14.7% and 9.0%). The superior F1 of the pipeline variant can be attributed to its notably higher precision. By narrowing down the candidate search space to fewer, more relevant files, the pipeline approach reduces noise and minimizes false positives, which directly improves precision and con-sequently boosts the F1 score.

Performance improves from Closed-Book to Project-Structure to Location-Hint. Pipeline strategies show mixed results: both weaken file-level accuracy but can improve precision–recall balance at the class and function levels by narrowing predictions to earlier-identified files. Overall, project context and file hints clearly enhance localization, while pipelines are not uniformly effective since accurate localization requires integrating information across levels rather than following staged processes. The details are as follows:

- **Closed-Book:** Performs the weakest. At the file, class, and function levels, the best Acc@5 reaches only 25.0%, 17.7%, and 16.1%, and the best F1 drops to 22%, 11%, and 7.0%. These results highlight the inherent difficulty of localization when LLMs rely solely on prior knowledge.

- **Project-Structure:** This approach performs better than the Closed-Book setting at the file and class levels, while being slightly weaker at the function level (with a gap ¡4%). At the file, class, and function levels, the best Acc@5 reaches 29.0%, 24.5%, and 12.9%, and the best F1 achieves 22.1%, 12.4%, and 5.8%, respectively. The result indicates that providing high-level project structure offers useful contextual cues for coarse-grained localization, but remains insufficient for fine-grained reasoning at the function level.

- **Location-Hint:** This approach achieves the best overall results, except for the function-level F1 where the gap compared with the best is small (¡2%). At the file, class, and function levels, the best Acc@5 reaches 35.0%, 24.5%, and 21.0%, and the best F1 achieves 23.4%, 11.6%, and 8.3%, respectively. The results show that explicit location hints substantially enhance model performance, especially for coarse-grained localization, while still providing moderate gains at finer granularities.

- **Pipeline-ProStructure:** Its effectiveness does not necessarily surpass that of Project-Structure for general LLMs such as GPT-4o-mini and Claude 3.5, but it shows clear advantages for reasoning-oriented LLMs such as DeepSeek-R1. At the file, class, and function levels, the best Acc@5 reaches 33.7%, 24.5%, and 10.2%, and the best F1 achieves 21.2%, 12.0%, and 8.0%, respec-tively. The result indicates that the benefits of pipeline strategy depend strongly on the LLMs' reasoning capability, with reasoning-focused LLMs better able to exploit pipeline strategy for improved localization.

- **Pipeline-Hint:** This approach is generally weaker than Location-Hint for most models, with the exception of Claude 3.5 at the class and function levels, where it performs comparably, partic-ularly on the F1 metric. At the file, class, and function levels, the best Acc@5 reaches 32.0%, 20.4%, and 12.9%, and the best F1 achieves 22.5%, 14.7%, and 9.0%, respectively. The result indicates that pipelining can introduce noise or error propagation that weakens the effectiveness of location hints, although stronger models such as Claude 3.5 are more robust to such pipeline effects.

A.7   LINE-LEVEL RESULTS OF STATE-OF-THE-ART APPROACHES ON FULL MULOCBENCH

Inspired by the prior fault localization works Li et al. (2019); Yang et al. (2024a); Kang et al. (2024), we do not report accuracy because developers do not need all predicted lines, and many lines can introduce noise. We evaluate Top-N metrics, where the prediction is considered correct if any ground-truth line appears in the top-N predicted lines. Table 8 shows the line-level compar-ison with state-of-art approaches on Python code localization in MULocBench. Table 8 shows the line-level comparison with prompting-LLM approaches on full MULocBench.

The result shows line-level localization is challenging: over 60% of issues fail to localize any correct line, and even the best SOTA method (DeepSeekR1) achieves only 35.5% Top@5 on Python code

| Type | LLM | Line(%) | |
|------|-----|---------|---|
| | | Top@1 | Top@5 |
| BM25 | | 2.0 | 4.5 |
| Agentless | GPT-4o-mini | 3.8 | 3.8 |
| | Claude3.5 | 7.5 | 9.2 |
| | DeepSeekR1 | 3.3 | 5.5 |
| LocAgent | GPT-4o-mini | 5.4 | 5.4 |
| | Claude3.5 | 16.1 | 23.7 |
| | DeepSeekR1 | 19.4 | 35.5 |
| OpenHands | GPT-4o-mini | 9.7 | 22.6 |
| | Claude3.5 | 17.2 | 21.5 |
| | DeepSeekR1 | 20.4 | 26.9 |

Table 7: Line-Level comparison with state-of-art approaches on full MULocBench

| Type | LLM | Line(%) | |
|------|-----|---------|---|
| | | Top@1 | Top@5 |
| ClosedBook | GPT-4o-mini | 6.5 | 8.4 |
| | Claude3.5 | 4.1 | 4.6 |
| | DeepSeekR1 | 6.0 | 9.6 |
| ProStructure | GPT-4o-mini | 7.0 | 8.4 |
| | Claude3.5 | 5.3 | 5.7 |
| | DeepSeekR1 | 3.6 | 6.0 |
| Hint | GPT-4o-mini | 7.5 | 9.6 |
| | Claude3.5 | 5.5 | 6.1 |
| | DeepSeekR1 | 4.8 | 10.8 |
| Pipeline-ProStructure | GPT-4o-mini | 3.9 | 5.2 |
| | Claude3.5 | 5.3 | 5.9 |
| | DeepSeekR1 | 2.4 | 6.0 |
| Pipeline-Hint | GPT-4o-mini | 5.2 | 6.0 |
| | Claude3.5 | 4.8 | 5.4 |
| | DeepSeekR1 | 4.8 | 9.6 |

Table 8: Line-Level comparison with LLM-based approaches on full MULocBench

localization. On full MULocBench, prompt-based LLMs perform substantially worse, with the best Top@5 reaching only around 10%.

## A.8 Temporal Analysis

Table 9 presents the file-, class- and function- level metrics across LLMs under the Closed-Book setting for 12 different temporal partitions that group issues by the years in which the issues were created. It is evident from the Table that there is no consistent correlation between model performance and year, supporting our conclusion that MULocBench is unlikely to suffer from temporal data leakage, despite models potentially having seen older issues within their pre-training datasets. The details are as follows:

- **Low Performance Across All Years**: Performance is consistently low, even at file-level, with the best Acc@5 remaining below 35% for each year. The result indicates that issue localization is different from simple copy-paste tasks; it requires semantic understanding and reasoning, and cannot be reliably achieved through memorization of data.

- **No Temporal Advantage**: Older issues do not show higher accuracy, indicating that models are not benefiting from potential memorization. For example, at the file-level, Claude 3.5 achieves 18.5% on issues from 2014 and earlier, 4.7% on issues from 2016 , and 13.5% on issues from 2020. In contrast, recent issues from 2025 reach 30.8%, despite being few in number and beyond any model's knowledge cutoff. These non-monotonic variations over time indicate that performance differences primarily reflect the inherent difficulty of issues rather than data memorization.

| Year | Issue Number | LLM | File(%) | | | | | Class(%) | | | | | Function(%) | | | | |
|---|---|---|---|---|---|---|---|---|---|---|---|---|---|---|---|---|---|
| | | | Acc@1 | Acc@5 | P | R | F1 | Acc@1 | Acc@5 | P | R | F1 | Acc@1 | Acc@5 | P | R | F1 |
| 2025 | 14 | Claude3.5 | 30.8 | 30.8 | 33.3 | 33.3 | 33.3 | 0.0 | 0.0 | 25.0 | 25.0 | 25.0 | 20.0 | 20.0 | 16.7 | 20.0 | 18.2 |
| 2024 | 161 | Claude3.5 | 13.1 | 15.0 | 19.8 | 11.7 | 14.7 | 6.8 | 6.8 | 11.4 | 4.5 | 6.5 | 6.5 | 6.5 | 8.3 | 3.6 | 5.0 |
| 2023 | 246 | Claude3.5 | 14.5 | 21.5 | 20.4 | 14.5 | 17.0 | 8.6 | 8.6 | 18.5 | 6.5 | 9.7 | 6.3 | 10.2 | 10.0 | 5.3 | 6.9 |
| 2022 | 112 | Claude3.5 | 17.8 | 27.1 | 25.8 | 15.6 | 19.5 | 14.8 | 16.4 | 29.5 | 7.4 | 11.9 | 11.4 | 11.4 | 15.9 | 6.5 | 9.2 |
| 2021 | 110 | Claude3.5 | 21.3 | 29.6 | 35.6 | 23.6 | 28.4 | 17.6 | 17.6 | 36.7 | 12.9 | 19.1 | 8.6 | 10.0 | 13.7 | 7.3 | 9.5 |
| 2020 | 95 | Claude3.5 | 10.1 | 13.5 | 29.6 | 15.6 | 20.5 | 7.1 | 7.1 | 22.2 | 7.0 | 10.7 | 10.2 | 12.7 | | 4.4 | 6.6 |
| 2019 | 79 | Claude3.5 | 15.4 | 25.6 | 26.1 | 17.9 | 21.2 | 8.9 | 8.9 | 15.2 | 6.8 | 9.4 | 11.1 | 13.0 | 5.7 | 3.0 | 3.9 |
| 2018 | 80 | Claude3.5 | 20.3 | 22.8 | 26.1 | 16.3 | 20.0 | 8.5 | 8.7 | 16.7 | 6.2 | 9.1 | 10.2 | 10.4 | | 4.9 | 6.6 |
| 2017 | 68 | Claude3.5 | 10.6 | 18.2 | 20.6 | 11.2 | 14.5 | 8.3 | 11.1 | 16.7 | 2.3 | 4.1 | 9.8 | 9.8 | 7.5 | 3.2 | 4.5 |
| 2016 | 47 | Claude3.5 | 2.3 | 4.7 | 26.2 | 16.3 | 20.1 | 0.0 | 3.3 | 25.7 | 11.7 | 16.1 | 5.4 | 8.1 | 13.3 | 7.0 | 9.2 |
| 2015 | 33 | Claude3.5 | 9.4 | 15.6 | 31.0 | 15.0 | 20.2 | 8.7 | 8.7 | 13.0 | 4.6 | 6.8 | 13.8 | 17.2 | 12.9 | 4.2 | 6.3 |
| ≤2014 | 55 | Claude3.5 | 14.8 | 18.5 | 28.3 | 18.9 | 22.6 | 5.6 | 5.6 | 16.3 | 8.0 | 10.7 | 7.5 | 10.0 | 10.1 | 6.4 | 7.8 |
| 2025 | 14 | GPT-4o-mini | 23.1 | 23.1 | 26.7 | 26.7 | 26.7 | 0.0 | 0.0 | 25.0 | 25.0 | 25.0 | 0.0 | 0.0 | 0.0 | 0.0 | 0.0 |
| 2024 | 161 | GPT-4o-mini | 10.0 | 13.8 | 9.7 | 6.6 | 7.8 | 5.5 | 6.8 | 5.0 | 3.5 | 4.1 | 3.2 | 5.4 | 2.4 | 2.2 | 2.3 |
| 2023 | 246 | GPT-4o-mini | 8.3 | 11.6 | 12.7 | 10.5 | 11.5 | 5.7 | 5.7 | 4.0 | 3.9 | 3.9 | 5.5 | 7.9 | 3.1 | 3.7 | 3.4 |
| 2022 | 112 | GPT-4o-mini | 15.0 | 21.5 | 17.9 | 12.5 | 14.7 | 8.2 | 9.8 | 8.7 | 5.1 | 6.5 | 10.0 | 11.4 | 6.0 | 5.0 | 5.5 |
| 2021 | 110 | GPT-4o-mini | 13.9 | 25.0 | 21.5 | 14.9 | 17.6 | 13.7 | 15.7 | 19.5 | 10.8 | 13.9 | 8.6 | 10.0 | 7.1 | 8.1 | 7.6 |
| 2020 | 95 | GPT-4o-mini | 7.9 | 10.1 | 18.8 | 12.7 | 15.1 | 9.5 | 9.5 | 14.1 | 12.3 | 13.1 | 8.5 | 11.9 | 6.7 | 5.6 | 6.1 |
| 2019 | 79 | GPT-4o-mini | 11.5 | 23.1 | 16.4 | 12.6 | 14.2 | 8.9 | 11.1 | 9.7 | 9.6 | 9.7 | 3.7 | 11.1 | 3.9 | 4.0 | 4.0 |
| 2018 | 80 | GPT-4o-mini | 17.7 | 20.3 | 17.3 | 11.5 | 13.8 | 6.5 | 6.5 | 5.2 | 4.7 | 4.9 | 5.1 | 8.5 | 3.5 | 4.0 | 3.7 |
| 2017 | 68 | GPT-4o-mini | 7.6 | 15.2 | 16.1 | 7.8 | 10.5 | 5.6 | 11.1 | 6.3 | 3.1 | 4.2 | 7.8 | 7.8 | 1.8 | 2.0 | 1.9 |
| 2016 | 47 | GPT-4o-mini | 4.7 | 9.3 | 21.6 | 14.7 | 17.5 | 13.3 | 13.3 | 20.3 | 18.2 | 19.2 | 2.7 | 10.8 | 6.0 | 7.6 | 6.7 |
| 2015 | 33 | GPT-4o-mini | 9.4 | 9.4 | 15.6 | 8.3 | 10.9 | 8.7 | 13.0 | 7.5 | 4.6 | 5.7 | 10.3 | 10.3 | 5.1 | 2.6 | 3.5 |
| ≤2014 | 55 | GPT-4o-mini | 13.0 | 25.9 | 22.1 | 15.7 | 18.4 | 5.6 | 8.3 | 11.2 | 10.0 | 10.6 | 5.0 | 12.5 | 4.6 | 7.0 | 5.6 |

Table 9: Performance comparison with Closed-Book across different years on the MULocBench

## A.9 Figures of Performance comparison of different issue types, reasons, location scopes and types on Full MULocBench

***Peformance Comparison across Issue Types.*** Figure 7 presents the performance comparison across different issue types on the full MULocBench. The performance differences observed on Python code closely mirror those on the full benchmark. **Execution Failures** achieve moderate results, with Location-Hint reaching 28% Acc@5 and 26% F1 at the file level. Their performance is relatively balanced across location levels (file, class, function) and metrics (Acc@1, Acc@5, precision, recall, F1), typically yielding mid-range scores: stronger than Enhancement Requests and Unexpected Results but weaker than Usage Questions, indicating a moderate level of localization difficulty. **Unexpected Results and Enhancement Requests** yield the weakest performance, with Acc@5 capped at around 20% at the file level. At the class level, Acc@5 and F1 are around 10% and 15%, while at the function level both fall below 10%, reflecting the challenge of pinpointing subtle or underspecified behavioral issues. **Usage Questions** achieve the strongest overall performance, with Location-Hint reaching 40% Acc@5 and 28% F1 at the file level. At the class level, Acc@5 and F1 are 29% and 27%, and at the function level both remain around 10%. Their relatively higher performance likely due to clearer textual cues in issue descriptions that make localization easier.

***Peformance Comparison across Issue Reasons.*** Figure 8 shows the performance comparison of different issue reasons on the full MULocBench. **Implementation Bug** achieve the most stable performance, with Location-Hint reaching 23% Acc@5 and 26% F1 at the file level, and 12–15% at finer levels. Although bug fixing is generally considered highly challenging, the relatively balanced performance across levels, together with the long-standing emphasis of researchers on bug-fix scenarios, explains why this issue reason performs more consistently than others. **Design Deficiencies** perform weakest, with best file-level results of 20% Acc@5 and 21% F1, but Acc@5 and F1 at the class and function levels drop to 5.4%–16%. Notably, precision is obviously higher than recall across all three levels. Across all levels, precision is higher than recall, as design-related issues are abstract and cross-cutting: they are harder to locate, but once identified, predictions tend to be precise. **User-Induced Problems** show intermediate performance, with 33% Acc@5 and 20% F1 at the file level, but only 7%–13% at the class and function levels. This suggests that user mistakes often provide clear textual cues that aid file-level localization. However, such cues are less informative for finer-grained levels, which results in weaker class- and function-level performance.

## A.10 Cross-Project Issue Localization Analysis

Cross-project issue localization remains extremely challenging: performance drops to 0%, even when Hint-LLM–based locations are provided, as shown in Figure 9. This indicates that cross-project issue localization requires a type of reasoning that current LLMs are particularly weak at.

Several factors contribute to this difficulty. First, issue descriptions rarely specify the responsible project explicitly. Second, the cross-project setting dramatically enlarges the search space compared

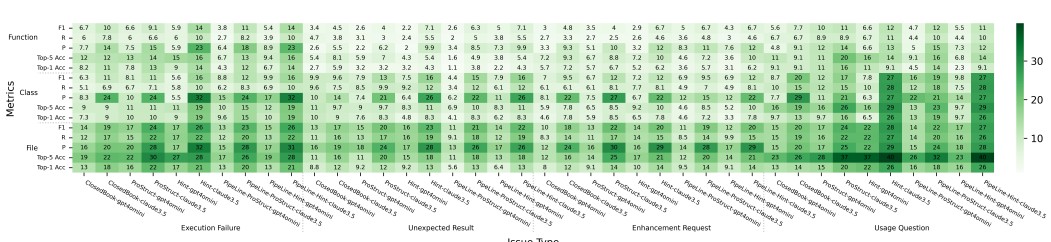

Figure 7: Performance comparison of different issue types on full MULocBench.

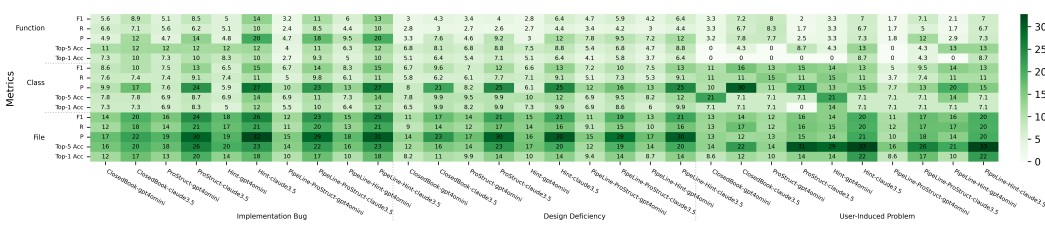

Figure 8: Performance comparison of different issue reasons on full MULocBench.

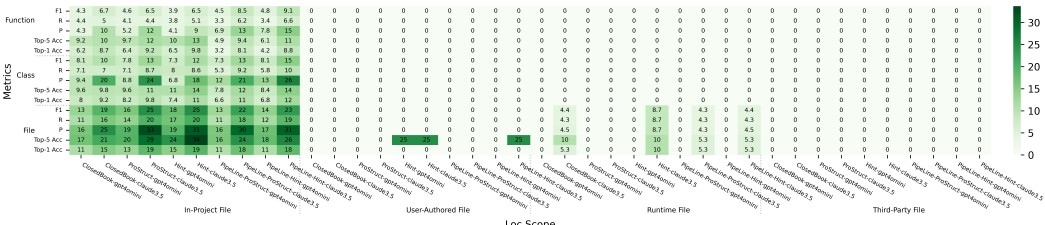

Figure 9: Performance comparison of different location scopes on full MULocBench.

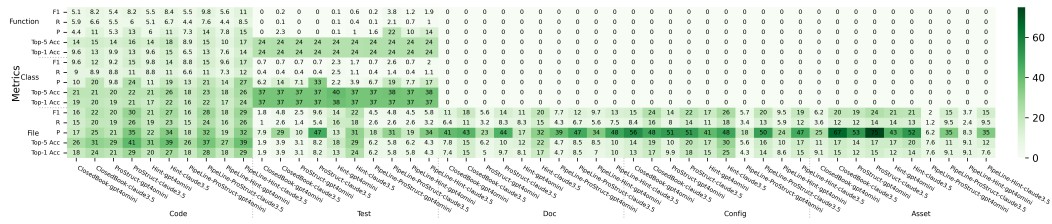

Figure 10: Performance comparison of different location types on full MULocBench.

with in-project localization. Third, cross-project issues are far fewer than in-project ones, inducing a strong in-project bias during prediction.

Consequently, although LLMs should perform deep cross-project reasoning, they tend to rely on shallow keyword matching between issue descriptions and project names, leading to poor localization accuracy. For example, for the issue "module torch.cuda has no attribute comm," LLMs repeatedly predict in-project files such as `torch/cuda/comm.py`. Even when explicitly prompted that the fix should be in a third-party project, the model gravitates toward projects with names similar to `torch` (e.g., `torch`, `torchvision`, `torchaudio`), whereas the correct project is actually `InterHand2.6M`, revealing the model's limited ability to reason across repositories.

Another example occurs in the issue "Chunk-encoded request doesn't recognize iter_content generator." LLMs typically respond with projects such as `requests`, `urllib3`, or `httplib`, simply because these names appear in the issue description. However, the true resolution lies in the `toolbelt` project, which must buffer generator-produced data before passing it to `requests`.

These examples demonstrate that without explicit cross-project supervision, LLMs default to in-project biases and keyword matching rather than performing genuine cross-repository reasoning.

### A.11 A QUALITATIVE FAILURE ANALYSIS OF ISSUE LOCALIZATION ON MULOCBENCH

We provide five qualitative analyses of wrong issue localization from state-of-the-art approaches or LLMs.

---

**Issue Description**   *pytorch/pytorch/issues/48435*

**AttributeError: module 'torch.cuda' has no attribute 'comm**

I'm using torch 1.7.0, and get this kind of error

my torch is installed via  pip install torch==1.7.0+cu101 torchvision==0.8.1+cu101 torchaudio===0.7.0
…

---

**LLM's  Response**

**GroundTruth**
Project: InterHand2.6M
File: common/nets/module.py
Function: soft_argmax_1d
Line: 43

Closed-Book DeepSeek-R1 ❌
Project: pytorch
File: **torch/cuda/__init__**.py
Line: 22
Line: 23

File: **torch/cuda/comm**.py
Line: 1
Line: ...
...

Pro-Structure GPT-4omini ❌
Project: pytorch
File: **torch/cuda**/__init__.py
Line: 1

Project: pytorch
File: **torch/cuda**/comm.py
Line: 1
...

Hint Claude3.5 ❌
Project: pytorch
File: **torch**/csrc/**cuda**/comm.cpp
File:
**torch**/csrc/**cuda**/Module.cpp
...
File:
**torch/cuda/__init__**.py
..

Pipeline-ProStructure DeepSeek-R1 ❌
Project: pytorch
File: **torch/cuda/__init__**.py
Line: 1

Project: pytorch
File: **torch/cuda**/comm.py
Line: 1
…

Pipeline-Hint GPT-4o-mini ❌
Project: pytorch
File: **torch**/csrc/**cuda**/comm.cpp
Line: 1
Line: 2
..
project: pytorch
file: **torch**/csrc/**cuda**/comm.h
...

---

**Explanation**

**Word-Matched Misleading:** LLMs rely on keywords such as torch, cuda, or comm in the issue description to search for files containing these terms as candidate locations, even when these files are unrelated to the actual root cause.

The part analysis of DeepSeek-R1 is as follows:
The error indicates that `torch.cuda.comm` is missing, which suggests issues with CUDA communication operations. The most important files to check are:
1. Core CUDA communication implementation files (`torch/cuda/comm.py`, `torch/csrc/cuda/comm.cpp`)
2. Module initialization files (`torch/cuda/__init__.py`, `torch/csrc/cuda/Module.cpp`) …

We also interacted with LLMs, such as GPT-4o-mini, to clarify that the issue originates from a third-party project. However, the model again suggested incorrect projects such as torch or torchvision, purely based on keyword matching.

Figure 11:  In this example, LLMs rely on keywords such as torch, cuda, or comm in the issue description to search for files containing these terms as candidate locations, even when these files are unrelated to the actual root cause.

## A.12   PROMPT TEMPLATES OF LLM-BASED APPROACHES

Figure 16, Figure 17, Figure 18, Figure 19, Figure 20, Figure 21, Figure 22 shows the prompt templates of LLM-based approaches.

---

**Issue Description**     *Illyasviel/Fooocus/issues/3689*

Exits upon attempting to load a model on Windows
**What happened?**
Attempting to run Fooocus on Windows 11 simply exits when attempting to load the default model, no error or nothing.
**Steps to reproduce the problem**
1.Install Fooocus on Windows 11 with a NVIDIA GPU ...
**What should have happened?**
It should've loaded the model successfully. ....

**LLM's Response**

**GroundTruth**
Project Name: Fooocus
File: presets/default**.json**
Line: 2

File: config**.txt**
File: config_modification_tutorial**.txt**

Closed-Book GPT-4o-mini
Project: Fooocus    ✖
File: src/main**.py**
Class: MainApp
Function: load_model
Line: 45
...
File: src/utils/model_loader**.py**
Class: ModelLoader
Function: load_default_model
Line: 30
...

Pro-Structure Claude3.5
Project: Fooocus
File: entry_with_update**.py**    ✖
Function: main
…
File: shared**.py**
Function: initialize_environment
…

Hint DeepSeek-R1    ✖
Project Name: Fooocus
File: modules/model_loader**.py**
Class: ModelLoader
Function: load_model
Function: load_default_model
Line: …
File: modules/core**.py**
Function: load_models
..

Pipeline-ProStructure GPT-4o-mini    ✖
Project: Fooocus
File: entry_with_update**.py**
Line: 1
…
File: backend/modules/model_loading**.py**
Line: 1
…

Pipeline-Hint GPT-4o-mini    ✖
project: Fooocus
File: modules/model_loader**.py**
Function: load_file_from_url
…
File: modules/core**.py**
Class: StableDiffusionModel
…

**Explanation**

**Code-Centric Bias:** LLMs incorrectly assume that the issue is code-related, resulting in all predicted locations pointing to Python source files.
LLMs then follow a series of reasoning steps based on this initial misjudgment, resulting in a list of Python files unrelated to the actual root cause. For example, part of the analysis produced by Claude 3.5 is as follows:
 Key areas to investigate:
1. Model loading logic → load_model function in entry_with_update.py
2. System compatibility checks → check_system_compatibility in modules/config.py
3. Path resolution → get_model_path function in modules/system_specific.py
4. Initialization procedures → initialize_model function in modules/core.py

Figure 12: In this example, LLMs mistakenly assume that the issue is code-related. Based on this incorrect assumption, they decompose the model-loading process step by step, generating a list of Python code locations that are unrelated to the actual failure.

| Issue Description | *pallets/flask/issues/5160* |
|---|---|

**Switch to importlib breaks scripts with app.run()**
Switch to importlib breaks scripts with `app.run()`With a trivial script:
if __name__ == "__main__":
app.run(debug=True)
…

**LLM's Response**

**GroundTruth**
Project: flask
File: src/flask/scaffold.py
Function: _find_package_path
Line: 825
Line: 826
…
File: **src/flask/helpers.py**
Function: get_root_path
..

Hint DeepSeek-R1 ✖
Project: flask
File: src/flask/scaffold.py
Function: _find_package_path
Line: 858
…
Function: find_package
File: src/flask/app.py
File: tests/test_basic.py
File: src/flask/__main__.py
…

Closed-Book GPT-4o-mini ✖
Project: flask
File: src/flask/app.py
Class: Flask
Function: __init__
...
project: flask
file: src/flask/scaffold.py
Function: find_package
Line: 898
Line: 858

Pipeline-ProStructure GPT-4o-mini ✖
Project: flask
File: src/flask/app.py
Line: 376
Line: 630
…
File: src/flask/scaffold.py
…

Pro-Structure Claude3.5 ✖
Project: flask
File: src/flask/scaffold.py
Function: _find_package_path
Line: 858
...
File: src/flask/app.py

Pipeline-Hint CLaude3.5 ✖
Project: pytorch
Project: flask
File: src/flask/scaffold.py
Function: _find_package_path
Line: 858
…
File: src/flask/app.py
…

**Explanation**

**Responsibility Chain Tracking Limitation:** LLMs can localize scaffold.py, where importlib is used to find package specs, but fail to identify **helpers.py**, which implements the fallback logic triggered by scaffold.py. Instead, they over-predict unrelated files like app.py or test modules.

The part analysis of Claude3.5 is as follows:
here are the key locations that need to be edited to fix the issue with importlib breaking scripts using `app.run()`:
**Explanation:**
1. The core fix needs to be in `scaffold.py` where importlib is used to find package specs
2. `app.py` needs updates since it calls the affected package finding functions
3. Tests must be added/updated to verify the fix works for `__main__` scripts
4. Documentation should be updated to reflect any changes
5. Example projects using `app.run()` should be verified to work correctly

Figure 13:  In this example, LLMs correctly identify scaffold.py, where importlib is used to find package specs, but fail to identify helpers.py, which implements the fallback logic triggered by scaffold.py. Instead, LLMs over-predict unrelated files such as app.py or test modules, illustrating a failure to trace the narrow responsibility chain.

| | Issue Description | *geekan/MetaGPT/issues/1174* |
|---|---|---|

**New provider: Amazon Bedrock (AWS)**
Feature description
Please include support for Amazon Bedrock models. These models can be from Amazon, Anthropic, AI21, Cohere, Mistral, or Meta Llama 2.
Your Feature
  Create a new LLM Provides under metagpt/provider for Amazon Bedrock
  Include it in the LLMType available

**LLM's Response**

**GroundTruth**
Project: MetaGPT
File: metagpt/configs/llm_config.py
Line: 19
Line: 23

File: metagpt/configs/llm_config.py
Class: LLMType
Line: 34

Class: LLMConfig
Line: 77
…
File: metagpt/utils/**token_counter.py**
Line: 212
File: metagpt/provider/__init__.py
File: tests/metagpt/provider/**mock_llm_config.py**
..

Agentless DeepSeek-R1 ❌
File:
metagpt/**provider/bedrock/bedrock_provider.py**
Class: MistralProvider
Line: 104
Line: 105
..
File: **metagpt/provider/bedrock**_api.py
…
File: **metagpt/provider/bedrock**/base_provider.py
Class: BaseBedrockProvider

Function: get_request_body
…

BM25 ❌
File: …**provider/bedrock/bedrock_provider**.py
Class: MistralProvider
Class: MetaProvider
…

LocAgent Claude3.5 ❌
File: metagpt/configs/llm_config.py # Add new enum for Amazon Bedrock
Class: **LLMType**
Line: 17
..
File: metagpt/**provider/bedrock.py** #Create new Amazon Bedrock LLM Provider implementation
Line: 1
…
File: metagpt/provider/__init__.py #Import and register the new Bedrock provider
File: metagpt/**llm**.py
…

OpenHands GPT-4o-mini ❌
File: metagpt/configs/llm_config.py # Add
Class: LLMType

Function: api_type
File: metagpt/provider/__init__.py.
File: **metagpt/provider**/llm_provider_registry.py
Function: **LLMProvide**rRegistry
…

**Explanation**

**Word-Matched Misleading:** SOTA approaches are misled by file paths containing keywords such as ``provider'' or ``bedrock'', even though these files are unrelated to the issue localization.
**Responsibility Chain Tracking Limitation:** Although SOTA approaches like LocAgent and OpenHands can identify obvious components such as configs/llm_config.py, they fail to capture dependency files that are critical for the feature's functionality and verification. For example, **token_counter.py** handles token counting for the new provider, and **mock_llm_config.py** provides the test scaffolding needed to validate its behavior.

Figure 14: In this example from the MetaGPT feature request, SOTA approaches such as LocAgent and OpenHands exhibit keyword matched pseudo localization by being misled by files containing terms like "provider" or "bedrock". They also show responsibility chain tracking failures by missing critical dependency files such as token_counter.py and mock_llm_config.py that are required for correct functionality and verification.

**Issue Description**  *ansible/ansible/issues/27729*

**Removed restricted key from module data: ansible_lxc_bridge**
COMPONENT NAME: **Gathering Facts**
ANSIBLE VERSION: ansible 2.3.1.0  config file = /etc/ansible/ansible.cfg
configured module search path = Default w/o overrides
OS / ENVIRONMENT Archlinux, but probably not platform specific
SUMMARY
During **gathering facts** I get following warning

TASK [**Gathering Facts**] *********************************************************************
 [WARNING]: **Removed restricted key from module data: ansible_lxc_bridge** = {u'macaddress':
u'70:85:c2:0b:a3:4a', u'features': {}, u'interfaces': [u'vethG18OR8', u'enp0s31f6',
u'vethWYJVBN'], u'mtu': 1500, u'active': True, u'promisc': False, u'stp': False, u'ipv4':
{u'broadcast': u'192.168.0.255', u'netmask': u'255.255.255.0', u'network': u'192.168.0.0',
u'address': u'192.168.0.110'}, u'ipv6': [{u'scope': u'link', u'prefix': u'64', u'address':
u'fe80::7285:c2ff:fe0b:a34a'}], u'device': u'lxc_bridge', u'type': u'bridge', u'id':
STEPS TO REPRODUCE: Try to run **playbook** against host with **lxc** and bridge configured.
EXPECTED RESULTS: No warning.

**LLM's  Response**

**GroundTruth**
Project: ansible
File: lib/ansible/playbook/task.py
Class: Task
Function: preprocess_data
Line: 208
Line: 214
…
File: lib/ansible/plugins/action/__init__.py
Class: ActionBase
Function: _clean_returned_data
Line: 783

Agentless DeepSeek-R1 ✖
**File: setup.py**
Class: BuildPyCommand
Class: BuildScriptsCommand
Class: InstallLibCommand
Class: InstallScriptsCommand
Class: SDistCommand

Function: _find_symlinks
Function: _cache_symlinks
Function: _maintain_symlinks
**…**

**File: v1/tests/TestVaultEditor.py**

**Note:** The two files are retrieved using **embedding-based similarity** to the issue description.

BM25 ✖
File: v1/ansible/inventory/__init__.py
Class: Inventory
Function: get_**restriction**
Function: **restrict**_to
…

LocAgent Claude3.5 ✖
File:
lib/ansible/module_utils/**facts**/network/linux.py
# otential location for network interface fact gathering
File: lib/ansible/module_utils/**facts/collector**.py
# potential location for fact collection logic
File: lib/ansible/modules/system/setup.py #
core module for gathering facts
File:
lib/ansible/module_utils/**facts**/namespace.py
…

OpenHands GPT-4o-mini ✖
File: lib/ansible/executor/**play_iterator.py**
Class: PlayIterator      Note: File instantiating the
Function: __init__       "Gathering Facts" task
…                        setup_task.name = 'Gathering Facts'
File: contrib/inventory/**lxc**_inventory.py
Class: **lxc**
File: contrib/inventory/libvirt_**lxc**.py
Class: libvirt_**lxc**
File: contrib/inventory/proxmox.py
…

**Explanation**

**Word-Matched Misleading:** SOTA approaches overfit to **keyword similarity** (e.g., "lxc", "bridge", "fact"), retrieving unrelated inventory or network modules. They miss the true root cause in regex-based sanitization logic that contains no matching keywords.
**Responsibility Chain Tracking Limitation:**  SOTA approaches fail to trace the responsibility chain across the multi-stage execution pipeline of fact gathering.
Although the issue arises during **"Gathering Facts,"** the relevant control flow spans:
**task preprocessing** (playbook/task.py: preprocess_data),
**action result sanitization** (action/__init__.py: _clean_returned_data), and
**regex-based filtering** of ansible_<conn>_* keys (re.compile('^ansible_%s_' % conn_name)).
Current systems cannot follow this chain and instead **attribute the failure to modules that merely**
***mention* LXC or network interface**s.

Figure 15:  In this example In this example, SOTA methods are misled by surface keyword matches (e.g., lxc, fact) and fail to trace the multi-stage responsibility chain behind fact gathering, causing them to localize the issue to irrelevant inventory or network modules instead of the true root cause in regex-based sanitization.

---

**Prompt Template of Closed-Book LLM**

Please look through the following GitHub problem description and provide a complete list of locations with full path that one would need to edit to fix the problem.

Each location should include:
Project name
Full file path (absolute within the repository, e.g., src/utils/helpers.py)
Class names (only for source code files)
function names (only for source code files)
Line numbers

Notes:
1. Only include the full file paths in your response.
2. Include all necessary files to resolve the issue, and do not include any unrelated files.
3. Your answer no more than 30 files.

The returned files should be separated by new lines ordered by most to least important and wrapped with ``````

### **GitHub Problem Description** ###
{issue report consisting of project name, issue title and issue description}

###

for example, Your response format should like that: (each block only has one file)
```
project: XXX
file: file1.py
Class: XXX
Function: XXX
Line: XXX
Line: XXX

project: XXX
file: b/file1.js
Class: XXX
Class: XXX
Function: XXX
Function: XXX
Line: XXX
Line: XXX
Line: XXX

project: XXX
file: a/xx/yy/README.md
Line: XXX
Line: XXX
...
```

Figure 16: Prompt template of closed-book LLM in project localization for issue resolution.

---

**Prompt Template of Project-Structure LLM**

Based on the provided **Project Structure**, please look through the following GitHub problem description and provide a complete list of locations with full path that one would need to edit to fix the problem.

Each location should include:
Project name
Full file path (absolute within the repository, e.g., src/utils/helpers.py)
Class names (only for source code files)
function names (only for source code files)
Line numbers

Notes:
1. Only include the full file paths in your response.
2. Include all necessary files to resolve the issue, and do not include any unrelated files.
3. Your answer no more than 30 files.

The returned files should be separated by new lines ordered by most to least important and wrapped with
`````

### **GitHub Problem Description** ###
{issue report consisting of project name, issue title and issue description}

###

### **Project Structure**###
{project structure  is a hierarchical representation of the directory tree, along with relative paths, file names and file extensions}

###

for example, Your response format should like that: (each block only has one file)
```
project: XXX
file: file1.py
Class: XXX
Function: XXX
Line: XXX
Line: XXX

project: XXX
file: b/file1.js
Class: XXX
Class: XXX
Function: XXX
Function: XXX
Line: XXX
Line: XXX
Line: XXX

project: XXX
file: a/xx/yy/README.md
Line: XXX
Line: XXX
...
```

---

Figure 17: Prompt template of Project-Structure LLM in project localization for issue resolution.

---

**Prompt Template of Location-Hint LLM**

Based on the provided **Location Hint and Project Structure**, please look through the following GitHub problem description and provide a complete list of locations with full path that one would need to edit to fix the problem.

Each location should include:
Project name
Full file path (absolute within the repository, e.g., src/utils/helpers.py)
Class names (only for source code files)
function names (only for source code files)
Line numbers

Notes:
1. Only include the full file paths in your response.
2. Include all necessary files to resolve the issue, and do not include any unrelated files.
3. Your answer no more than 30 files.

The returned files should be separated by new lines ordered by most to least important and wrapped with
``````

### **GitHub Problem Description** ###
{issue report consisting of project name, issue title and issue description}

###

### **Location Hint**###
Your found Locations should be { in-project file/runtime file/third-party file/user-authored file }
Your found file types should be code/test/documentation/configuration/asset }
###

### **Project Structure**###
{project structure is a hierarchical representation of the directory tree, along with relative paths, file names and file extensions. Note: According to the given location hint, only the corresponding file types are shown in the structure.
}

###

for example, Your response format should like that: (each block only has one file)
```
project: XXX
file: file1.py
Class: XXX
Function: XXX
Line: XXX
Line: XXX

project: XXX
file: b/file1.js
Class: XXX
Class: XXX
Function: XXX
Function: XXX
Line: XXX
Line: XXX
Line: XXX

project: XXX
file: a/xx/yy/README.md
Line: XXX
Line: XXX

...
```

Figure 18: Prompt template of Location-Hint LLM in project localization for issue resolution.

---

**Step 1 Prompt Template of Pipeline (Project-Structure Guided) LLM**

Based on the provided **Project Structure**, please look through the following GitHub problem description and provide a complete location list with full path that one would need to edit to fix the problem.

Each location should include:
Project name
Full file path (absolute within the repository, e.g., src/utils/helpers.py)

Notes:
1. Only include the full file paths in your response.
2. Include all necessary files to resolve the issue, and do not include any unrelated files.
3. Your answer no more than 30 files.

The returned files should be separated by new lines ordered by most to least important and wrapped with
` ` ` ` ` `

### **GitHub Problem Description** ###
{issue report consisting of project name, issue title and issue description}
###

### **Project Structure**###
{project structure is a hierarchical representation of the directory tree, along with relative paths, file names and file extensions.
}
###

for example, Your response format should like that: (each block only has one file)
` ` `
project: XXX
file: file1.py

project: XXX
file: b/file1.js

project: XXX
file: a/xx/yy/README.md
...
` ` `

Figure 19: Step 1 Prompt template of PipeLine (Project-Structure Guided) LLM in project localization for issue resolution.

---

**Step 2 Prompt Template of Pipeline (Project-Structure Guided) LLM**

Based on the **file list**, please look through the following GitHub problem description and provide a complete location list with full path that one would need to edit to fix the problem.

Each location should include:
Full file path (absolute within the repository, e.g., src/utils/helpers.py)
Class names (only for source code files)
function names (only for source code files)
Line numbers

Notes:
1. Only include the full file paths in your response.
2. Include all necessary files to resolve the issue, and do not include any unrelated files.
3. Your answer no more than 30 files.

Returned files should be separated by new lines ordered by most to least important and wrapped with ``````

### **GitHub Problem Description** ###
{issue report consisting of project name, issue title and issue description}
###

### **File List** ###
{{Files predicted by LLMs in Step 1. For valid source files, we also include their classes and functions extracted by parsing.}
###

for example, Your response format should like that: (each block only has one file)
```
project: XXX
file: file1.py
Class: XXX
Function: XXX
Line: XXX
Line: XXX

project: XXX
file: b/file1.js
Class: XXX
Class: XXX
Function: XXX
Function: XXX
Line: XXX
Line: XXX
Line: XXX

project: XXX
file: a/xx/yy/README.md
Line: XXX
Line: XXX
...
```

---

Figure 20: Step 2 Prompt template of PipeLine (Project-Structure Guided) LLM in project localization for issue resolution.

---

**Step 1 Prompt Template of Pipeline (Location-Hint Guided) LLM**

Based on the provided **Location Hint and Project Structure**, please look through the following GitHub problem description and provide a complete location list with full path that one would need to edit to fix the problem.

Each location should include:
Project name
Full file path (absolute within the repository, e.g., src/utils/helpers.py)

Notes:
1. Only include the full file paths in your response.
2. Include all necessary files to resolve the issue, and do not include any unrelated files.
3. Your answer no more than 30 files.

The returned files should be separated by new lines ordered by most to least important and wrapped with ` ` ` ` ` `

### **GitHub Problem Description** ###
{issue report consisting of project name, issue title and issue description}
###

### **Location Hint**###
Your found Locations should be { in-project file/runtime file/third-party file/user-authored file }
Your found file types should be code/test/documentation/configuration/asset }
###

### **Project Structure**###
{project structure is a hierarchical representation of the directory tree, along with relative paths, file names and file extensions. Note: According to the given location hint, only the corresponding file types are shown in the structure.
}
###

for example, Your response format should like that: (each block only has one file)
` ` `
project: XXX
file: file1.py

project: XXX
file: b/file1.js

project: XXX
file: a/xx/yy/README.md
...
` ` `

Figure 21: Step 1 Prompt template of PipeLine (Location-Hint Guided) LLM in project localization for issue resolution.

> **Step 2 Prompt Template of Pipeline (Location-Hint Guided) LLM**
>
> Based on the **file list**, please look through the following GitHub problem description and provide a complete location list with full path that one would need to edit to fix the problem.
>
> Each location should include:
> Full file path (absolute within the repository, e.g., src/utils/helpers.py)
> Class names (only for source code files)
> function names (only for source code files)
> Line numbers
>
> Notes:
> 1. Only include the full file paths in your response.
> 2. Include all necessary files to resolve the issue, and do not include any unrelated files.
> 3. Your answer no more than 30 files.
>
> Returned files should be separated by new lines ordered by most to least important and wrapped with ``````
>
> ### **GitHub Problem Description** ###
> {issue report consisting of project name, issue title and issue description}
> ###
>
> ### **File List** ###
> {{Files predicted by LLMs in Step 1. For valid source files, we also include their classes and functions extracted by parsing.}
> ###
>
> for example, Your response format should like that: (each block only has one file)
> ```
> project: XXX
> file: file1.py
> Class: XXX
> Function: XXX
> Line: XXX
> Line: XXX
>
> project: XXX
> file: b/file1.js
> Class: XXX
> Class: XXX
> Function: XXX
> Function: XXX
> Line: XXX
> Line: XXX
> Line: XXX
>
> project: XXX
> file: a/xx/yy/README.md
> Line: XXX
> Line: XXX
> ...
> ```

Figure 22: Step 2 Prompt template of PipeLine (Location-Hint Guided) LLM in project localization for issue resolution.

