# OpenReview forum: "MULocBench: Beyond Code Benchmark for Multi-Type Software Issue Localization"
_ICLR.cc/2026/Conference — ICLR 2026 Conference Withdrawn Submission_

### Official Review · Reviewer_QzsC · 2025-10-27

**Soundness:** 2
**Presentation:** 3
**Contribution:** 2
**Rating:** 2
**Confidence:** 4

**Summary:**

The paper presents MULocBench, a new dataset for evaluating issue localization in software projects. The authors argue that prior benchmarks are too narrowly focused on code-only bug fixes. In contrast, their benchmark includes 1,100 issues from 46 Python projects, featuring a wider variety of issue types. The authors use this benchmark to evaluate several existing state-of-the-art localization tools as well as five simple LLM prompting strategies. Their main reported finding is that all evaluated methods perform poorly, suggesting that realistic issue localization remains a significant open challenge.

**Strengths:**

1. The paper does an excellent job of articulating the limitations of existing benchmarks. The argument for a more diverse and realistic benchmark that reflects real-world software engineering practices is clear, compelling, and well-supported.
2. The paper is exceptionally well-written, clearly structured, and easy to follow. Figures and tables are informative and effectively communicate the key results and comparisons.

**Weaknesses:**

1. While the proposed benchmark and the accompanying evaluation are valuable, the paper's contribution is primarily empirical rather than methodological. It does not introduce a new technique to address the challenges it highlights. This focus on benchmarking, while important, may limit the paper's methodological novelty.
2. A major weakness is the lack of deep analysis into the failure modes of the evaluated models. The paper compellingly shows that current models fail, but it stops short of providing generalizable insights into why they fail. A qualitative analysis of specific failure cases would have been invaluable. For instance, are the failures due to long-context reasoning, misunderstanding of code semantics, or ambiguity in the issue descriptions?
A useful point of comparison is the well-received SWE-bench (ICLR 2024). However, its contribution was strengthened by a large, realistic, challenging and novel benchmark. And addressing these challenges is poised to drive significant innovations at both the applied and fundamental levels.
Without such insights, it is difficult for ML researchers to draw actionable conclusions for developing better models. While this is crucial work, its primary audience might be at top ESE venues (e.g., ICSE, FSE, ASE), where such benchmark and task-definition papers are highly impactful. I am open to raising my score if the authors' rebuttal can convincingly address the major concerns outlined above.

**Questions:**

1. Regarding the "location clarity" check mentioned around line 613: was this assessment performed manually by human annotators, or was an automated method (e.g., using an LLM) employed?
2. To better understand the nature of the challenge, could the authors provide a qualitative analysis of a few representative failure cases?

---

> ### Author Response · Authors · 2025-11-20
>
> >W2. A major weakness is the lack of deep analysis into the failure modes of the evaluated models.  .... While this is crucial work, its primary audience might be at top ESE venues (e.g., ICSE, FSE, ASE), where such benchmark and task-definition papers are highly impactful. I am open to raising my score if the authors' rebuttal can convincingly address the major concerns outlined above.
>
> >Q2. To better understand the nature of the challenge, could the authors provide a qualitative analysis of a few representative failure cases?
>
>
> **Failure analysis:**  Thanks for your suggestion! We select failure cases where LLMs or state-of-art methods failed. We have included the detailed description in Section 7 and five representative failed cases in Appendix A.11.
> We summarize three reasons as follows:
>
>  * **Word-Matching Misleading:** LLMs are misled by unrelated keywords in issue description overlaps with project file, class, or function names. For example, in Figure 11, the issue mentions torch.cuda, which appears in multiple files. Consequently, the model predicts files containing these keywords rather than the actual file, module.py, where the neural network functionality invoking torch.cuda is implemented.
>
>  * **Code-Centric Bias:** LLMs tend to assume that issues correspond to code-related problems. This assumption drives reasoning in the wrong direction.  For example,  in Figure 12, the issue states “load model failed,” but LLMs suggest code-related files, whereas the actual problem lies in configuration-related files.
>
>  * **Responsibility Chain Tracking Limitation:** LLMs struggle to trace multi-step responsibility chains, frequently predicting heuristically associated files instead of the critical downstream files. For example,  in Figure 13,  the issue  states “Switch to importlib break”,  LLMs correctly identify scaffold.py but miss the helper.py where implements the fallback logic triggered by scaffold.py.  LLMs over-predict files such as app.py, due to the fact that app.py often interacts with core modules like scaffold.py.
>
> **Benchmarking research aligns closely with ICLR’s scope and interests.**  ICLR embraces benchmarks across various domains, such as NutriBench (healthcare-ICLR25), BigCodeBench (software engineering-ICLR25), and SOO-Bench (optimization-ICLR25).  Going beyond traditional bug-fix-code issues, MULocBench reflects the realistic software issues across diverse issues and locations, and provides a reliable evaluation environment that exposes the limitations of existing methods and drives future methodological progress.
>
>
>
> >W1: While the proposed benchmark and the accompanying evaluation are valuable, the paper's contribution is primarily empirical rather than methodological. It does not introduce a new technique to address the challenges it highlights. This focus on benchmarking, while important, may limit the paper's methodological novelty.
>
> **Lack of new technique to address challenges in the benchmark:**
> Thanks for your comment! As prior popular benchmarks with GLUE(NLU), WMT(machine translation), NutriBench(healthcare), SOO-Bench(optimization), and SWEBench / BigCodebench(Software engineering), we would like to clarify that a benchmark is **not required** to introduce new techniques. The creation of a challenging and reliable benchmark is itself a fundamental driver of research/technique progress.
>
> Based on the previous failure analysis, we further propose possible suggestions for researchers to improve issue localization.
>
> * **Unrelated keywords masking.** Masking words reduces spurious lexical overlaps between issue descriptions and project locations(e.g., file names) so that LLMs can rely more on semantic cues rather than shallow word matching.
> * **File-type–aware keyword mapping.** Construct mapping libraries that associate each file type with representative keywords.These mappings can be used as a part of prompt to alleviate code-centric bias and guide models toward non-code or mixed-context issues.
> * **Project-level responsibility chains construction** Construct project-level responsibility chains that explicitly encode relationships among entities (project/file/class/function) to guide LLMs to perform deterministic deep reasoning.
>
>
> >Q1. Regarding the "location clarity" check mentioned around line 613: was this assessment performed manually by human annotators, or was an automated method (e.g., using an LLM) employed?
>
> **Location Clarity Check:** Thanks for your comment! Four annotators independently record resolution comments and the referenced locations, and any discrepancies from resolution comments or locations are resolved through discussion.  A location in resolved comments is labeled as clear if it is searchable in the project. Otherwise, it is labeled as unclear. For example, in an issue whose resolution comment is ``got it working, just changed ... in ui.py'',  and the ui.py is in the project, so the location is clear. We have included in Appendix A.1.

---

> > ### Comment · Reviewer_QzsC · 2025-11-27
> >
> > Thank you for your response; it has clarified some of my confusion.
> > But I'm still wondering why nearly all models do so poorly on MULocBench, yet perform relatively well on SWE-bench, even though both involve issue localization. Could this be attributed to the datasets not being identically distributed?

---

> > > ### Author Response · Authors · 2025-11-27
> > >
> > > Thanks for your comment! Yes, the data distributions differ substantially. Section 4 provides a detailed benchmark comparison, and Appendix A.2 reports full MULocBench statistics. Below we highlight the key differences:
> > >
> > > * **Beyond Python code**.  SWE-Bench Lite contains only Python code locations. In contrast, MULocBench covers a much broader set of file types, reflecting the heterogeneity of real-world issue locations. The details are as follows:
> > > | File Type | Issue number | File number |
> > > |-----------|-------------|------------|
> > > | Python    | 842         | 2494       |
> > > | OtherPL (JavaScript/C/Java)  | 16          | 53         |
> > > | Test  | 273         | 513        |
> > > | Doc  | 259        | 443        |
> > > | Asset  | 48        | 59        |
> > > | Configuration  | 167         | 308        |
> > >
> > > * **Mutliple files**. SWE-Bench Lite associates each issue with exactly one Python code file. MULocBench issues, however, span an average of 2.85 files (up to 30), which captures the multi-file debugging and investigation patterns commonly seen in practice.
> > >
> > > * **Broader issue types**. SWE-Bench Lite is heavily bug-fix–centric (≈77% are bug fixes), with the remaining being design deficiency. In contrast, MULocBench provides a balanced distribution across three issue types — bug (380), deficiency (399), and user-induced issues (321), which better reflects the diversity of real-world software problems.
> > >
> > > * **Beyond in-project files.** MULocBench goes beyond in-project files. It includes 80 issues whose target locations lie in runtime-generated files, third-party library files, or user-authored files, capturing scenarios that SWE-Bench Lite does not cover at all.
> > >
> > > In summary, SWE-Bench Lite represents a narrow slice of issue-localization scenarios, focusing primarily on single-file Python bug fixes within project boundaries. MULocBench, by contrast, captures the full complexity of real-world issue localization: heterogeneous file types, multi-file issue scopes, a balanced mix of issue types, and locations that extend beyond project code.

---

> > > > ### Comment · Reviewer_QzsC · 2025-11-28
> > > >
> > > > Thank you for your detailed rebuttal, which has addressed most of my concerns.
> > > >
> > > > I believe that bringing traditional software engineering (SE) tasks into the AI domain is a valuable direction that expands the scope of ICLR. I would be interested to hear your perspective on the future of this interdisciplinary area and the key challenges you foresee.
> > > >
> > > > Accordingly, I will be raising my evaluation score.
> > > >
> > > > As a minor point of feedback, you might consider highlighting the updated sections in the revised manuscript (e.g., in red or another color). This would greatly help reviewers in tracking the changes you've made.

---

> > > > > ### Author Response · Authors · 2025-11-28
> > > > >
> > > > > Thank you for your thoughtful comment and for your encouraging insights on the interdisciplinary direction between SE and the AI domain.
> > > > >
> > > > > We foresee a key future direction in advancing **AI’s deep and efficient reasoning abilities to understand complex systems**. Software provides an excellent testbed because software is not isolated: it consists of lots of files spanning diverse types, exhibits complex interactions among internal files, and interacts extensively with external libraries.
> > > > >
> > > > > One challenge lies in constructing and managing a unified **graph representation of the relationships** among code files, between code and non-code artifacts (e.g., configuration, documentation, tests), and with external dependencies. Another challenge is that, given the size and connectivity of these relational graphs, designing **effective search and reasoning strategies** is crucial to avoid combinatorial explosion and ensure both efficiency and correctness.
> > > > >
> > > > >
> > > > > We appreciate your suggestions and have highlighted the updated sections in red in the revised manuscript to help reviewers easily track the changes.

---

### Official Review · Reviewer_KfGY · 2025-10-27

**Soundness:** 3
**Presentation:** 3
**Contribution:** 2
**Rating:** 4
**Confidence:** 2

**Summary:**

This paper introduces MULocBench, a new benchmark for software issue localization. The authors argue that existing benchmarks, such as SWE-Bench and LocBench, are too narrowly focused on bug fixes resolved via pull requests. To address this limitation, they have constructed MULocBench, a diverse dataset of 1100 issues from 46 popular Python projects. The paper then reports on an extensive empirical study evaluating both state-of-the-art specialized tools and several LLM-based prompting strategies on this new benchmark. The central finding is that all evaluated methods perform poorly, with even the top-performing approaches failing to exceed 40% Acc@5 at the file level. This highlights a significant gap between current model capabilities and the demands of realistic software maintenance tasks.

**Strengths:**

1.  The primary contribution, the MULocBench dataset, is a significant asset for the community. The paper details a systematic methodology for data collection, filtering, and annotation. The public release of this diverse and realistic benchmark is a welcome contribution that will undoubtedly facilitate future research in this domain.
2. The experiments are thorough and well-designed. The authors evaluate a wide range of modern techniques, from traditional IR-based methods to sophisticated LLM agents.

**Weaknesses:**

The paper's primary weakness lies in its lack of analytical depth. While the low performance scores compellingly demonstrate that current models fail, the paper offers very little insight into why they fail. The analysis remains largely descriptive, functioning more as a report of performance metrics than a scientific investigation into the root causes of failure.

**Questions:**

1. The decision to exclude line-level evaluation due to low accuracy (<1%) seems premature. This low performance is, in itself, a highly significant finding that underscores the difficulty of the task. Could you elaborate on why function-level localization is considered "sufficient for developers,"(Line 288) given that practical fixes often require pinpointing a specific line or a small block of code within a function?
2. To ensure the benchmark is free from temporal data leakage (i.e., issues being part of the models' training data), could you provide more details on the date range (e.g., creation or resolution dates) for the issues included in MULocBench?
3. The LLM landscape evolves rapidly. Have you considered evaluating more recent or capable models, such as Reasoning Model (Deepseek-R1, claude-thinking)?

---

> ### Author Response · Authors · 2025-11-20
>
> > W1:The paper's primary weakness lies in its lack of analytical depth. While the low performance scores compellingly demonstrate that current models fail, the paper offers very little insight into why they fail. The analysis remains largely descriptive, functioning more as a report of performance metrics than a scientific investigation into the root causes of failure.
>
> **Failure analysis:**  Thanks for your suggestions! We select failure cases where LLMs or state-of-art methods failed. We have included the detailed description in Section 7 and five representative failed cases in Appendix A.11. We summarize three reasons as follows:
>
>  * **Word-Matching Misleading:** LLMs are misled by unrelated keywords in issue description overlaps with project file, class, or function names. For example, in Figure 11, the issue mentions torch.cuda, which appears in multiple files. Consequently, the model predicts files containing these keywords rather than the actual file, module.py, where the neural network functionality invoking torch.cuda is implemented.
>
>  * **Code-Centric Bias:** LLMs tend to assume that issues correspond to code-related problems. This assumption drives reasoning in the wrong direction.  For example,  in Figure 12, the issue states “load model failed,” but LLMs suggest code-related files, whereas the actual problem lies in configuration-related files.
>
>  * **Responsibility Chain Tracking Limitation:** LLMs struggle to trace multi-step responsibility chains, frequently predicting heuristically associated files instead of the critical downstream files. For example,  in Figure 13,  the issue  states “Switch to importlib break”,  LLMs correctly identify scaffold.py but miss the helper.py where implements the fallback logic triggered by scaffold.py.  LLMs over-predict files such as app.py, due to the fact that app.py often interacts with core modules like scaffold.py.
>
>
> > RQ2 To ensure the benchmark is free from temporal data leakage (i.e., issues being part of the models' training data), could you provide more details on the date range (e.g., creation or resolution dates) for the issues included in MULocBench?
>
> **Issue Date Details and Temporal Analysis** Thanks for your comments! The issue creation dates in MULocBench span from **2000-07-31** to **2025-03-02**. For clarity, we provide a temporal breakdown of issue counts by creation year and the corresponding LLM results in the Closed-Book setting in Table 9 of Appendix A.8 (excerpt below). For simplicity, we only show File Acc@5 here. The results suggest that MULocBench is unlikely to suffer from temporal data contamination.
> *  **Low Performance Across All Years**: Performance at the File level is consistently low, with Acc@5 remaining below 35% for all years, indicating negligible temporal data leakage.
> * **No Temporal Advantage**: Old issues do not show high accuracy, indicating that models are not benefiting from potential memorization. For example, for the File Acc@5 metric, Claude 3.5 achieves 18.5% on issues from 2014 and earlier, only 4.7% on issues from 2016 , and 13.5% on issues from 2020. In contrast, recent issues from 2025 reach 30.8%, despite being few in number and beyond any model’s knowledge cutoff. The non-monotonic fluctuations contradict the memorization hypothesis and instead reflect natural differences in issue difficulty. The result has been included in Appendix A.8.
> | Year   | Issue Number | Claude3.5 | GPT-4o-mini |
> |--------|-------------|-----------|-------------|
> | 2025   | 14          | 30.8      | 23.1        |
> | 2024   | 161         | 15.0      | 13.8        |
> | 2023   | 246         | 21.5      | 11.6        |
> | 2022   | 112         | 27.1      | 21.5        |
> | 2021   | 110         | 29.6      | 25.0        |
> | 2020   | 95          | 13.5      | 10.1        |
> | 2019   | 79          | 25.6      | 23.1        |
> | 2018   | 80          | 22.8      | 20.3        |
> | 2017   | 68          | 18.2      | 15.2        |
> | 2016   | 47          | 4.7       | 9.3         |
> | 2015   | 33          | 15.6      | 9.4         |
> | ≤2014  | 55          | 18.5      | 25.9        |

---

> ### Author Response · Authors · 2025-11-20
>
> > RQ1 The decision to exclude line-level evaluation due to low accuracy (<1%) seems premature. This low performance is, in itself, a highly significant finding that underscores the difficulty of the task. Could you elaborate on why function-level localization is considered "sufficient for developers,"(Line 288) given that practical fixes often require pinpointing a specific line or a small block of code within a function?
>
> Thanks for your comment! Following prior issue localization techniques such as LocAgent and CoSIL, function-level granularity is the finest evaluation level. Prior studies showing method level is the favored granularity level among developers[1], where it is efficient for later debugging or program repair[1,2,3].
>
> We acknowledge the value of line-level evaluation. Inspired by the prior fault localization works [2,4,5], we do not report accuracy because developers do not need all predicted lines, and many lines can introduce noise. We evaluate Top-N metrics, where the prediction is considered correct if any ground-truth line appears in the top-N predicted lines. The result shows  Line-level localization is challenging: over 60% of issues fail to localize any correct line, and even the best SOTA method (DeepSeekR1) achieves only 35.5% Top@5 on Python code. Prompt-based LLMs perform worse, with the best Top@5 around 10% on the full MULocBench. For simplicity, we excerpt a subset of approach results. The completed results have been included Tables 7 and 8 in Appendix A.7.
>
> | Type       | LLM           | Top@1 (%) | Top@5 (%) |
> |------------|---------------|-----------|-----------|
> | Agentless  | GPT-4o-mini   | 3.8       | 3.8       |
> |            | Claude3.5     | 7.5       | 9.2       |
> |            | DeepSeekR1    | 3.3       | 5.5       |
> | LocAgent   | GPT-4o-mini   | 5.4       | 5.4       |
> |            | Claude3.5     | 16.1      | 23.7      |
> |            | DeepSeekR1    | 19.4      | 35.5      |
> | OpenHands  | GPT-4o-mini   | 9.7       | 22.6      |
> |            | Claude3.5     | 17.2      | 21.5      |
> |            | DeepSeekR1    | 20.4      | 26.9      |
>
>
>
> | Type                   | LLM             | Top@1 (%) | Top@5 (%) |
> |------------------------|-----------------|-----------|-----------|
> | ClosedBook             | GPT-4o-mini | 6.5       | 8.4       |
> |                        | Claude3.5   | 4.1       | 4.6       |
> |                        | DeepSeekR1      | 6.0       | 9.6       |
> | Hint                   | GPT-4o-mini | 7.5       | 9.6       |
> |                        | Claude3.5l   | 5.5       | 6.1       |
> |                        | DeepSeekR1      | 4.8       | 10.8      |
> | Pipeline-Hint          | GPT-4o-mini | 5.2       | 6.0       |
> |                        | Claude3.5   | 4.8       | 5.4       |
> |                        | DeepSeekR1      | 4.8       | 9.6       |
>
>
> [1] Kochhar et al. Practitioners' expectations on automated fault localization ISSTA’16
>
> [2] Kang et al. A Quantitative and Qualitative Evaluation of LLM-Based Explainable Fault Localization FSE’24
>
> [3] Jiang et al. Impact of Code Language Models on Automated Program Repair ICSE’23
>
> [4]Tsumita et al. Large-scale evaluation of method-level bug localization with FinerBench4BL. SANER’23
>
> [5] Yang et al. Large Language Models for Test-Free Fault Localization ICSE’24

---

> ### Author Response · Authors · 2025-11-20
>
> > RQ3 The LLM landscape evolves rapidly. Have you considered evaluating more recent or capable models, such as Reasoning Model (Deepseek-R1, claude-thinking)?
>
> Thank you for your suggestion! We have incorporated DeepSeek-R1 into our experiments (see Table 1, Table 2, Table 7, and Table 8). Due to space constraints, we present here only the Acc@1 and Acc@5 results at the file, class, and method levels for LocAgent, closed-book, and Hint approaches.
>
> * At the file, class, and method levels, DeepSeekR1 outperforms general LLMs in LLM-prompt baselines, but it is not consistently superior in overall SOTA comparisons—for example, LocAgents' file-level Acc@5 is 29%, below Claude-3.5’s 35%, indicating **the limited marginal benefit of the reasoning model**.
>
> * At the line level, DeepSeekR1 outperforms general LLMs in LLM-prompt baselines and SOTA, indicating that **deep reasoning is particularly valuable for fine-grained localization**.
>
> | Type | LLM | File Acc@1 | File Acc@5 | Class Acc@1 | Class Acc@5 | Function Acc@1 | Function Acc@5 |
> |------|-----|------------|------------|-------------|-------------|---------------|---------------|
> | LocAgent | GPT-4o-mini | 11.7 | 16.1 | 6.6 | 10.8 | 4.3 | 9.4 |
> | LocAgent | Claude-3.5 | 28.5 | 35.2 | 17.6 | 25.9 | 14.9 | 21.9 |
> | LocAgent | DeepSeekR1 | 21.0 | 29.0 | 27.9 | 28.2 | 15.2 | 25.0 |
>
> | Type | LLM | File Acc@1 | File Acc@5 | Class Acc@1 | Class Acc@5 | Function Acc@1 | Function Acc@5 |
> |------|-----|------------|------------|-------------|-------------|---------------|---------------|
> | ClosedBook | GPT-4o-mini | 10.4 | 15.1 | 6.8 | 7.9 | 6.1 | 8.6 |
> | ClosedBook | Claude3.5 | 14.5 | 20.0 | 8.9 | 9.5 | 8.6 | 10.2 |
> | ClosedBook | DeepSeekR1-285 | 15.0 | 25.0 | 14.3 | 16.3 | 6.5 | 16.1 |
> | Hint | GPT-4o-mini | 13.5 | 22.3 | 6.8 | 10.1 | 6.5 | 9.9 |
> | Hint | Claude3.5 | 18.6 | 32.5 | 11.2 | 14.1 | 9.8 | 13.1 |
> | Hint | DeepSeekR1 | 18.0 | 35.0 | 16.3 | 24.5 | 16.1 | 21.0 |

---

### Official Review · Reviewer_5shn · 2025-11-01

**Soundness:** 3
**Presentation:** 3
**Contribution:** 3
**Rating:** 6
**Confidence:** 4

**Summary:**

The authors observe that existing localisation or SE benchmarks for LLMs restrict code localisation tasks to source-code locations, especially in the context of bug fixing. They remark that this does not cover the diversity of tasks in a software project, and propose to create a new benchmark that has a diversity of tasks and locations. They propose MULocBench and form this benchmark by mining issues from the top 50 most-starred GitHub projects. They filter to those issues that have clear resolutions and extract the location and associated metadata. Crucially, they allow location references to be Project Files, Runtime Files, configurations, user-authored files, etc.
To demonstrate the need for the new benchmark, the authors consider existing approaches validated on previous benchmarks, and re-evaluate them in terms of Acc(@1,@5), and P, R, F1(@inf?), showing that performance is lower at the file, class, and function levels.
To allow applying previous approaches, the result was restricted to the subset of the benchmark that targets source-code files.
To evaluate the full benchmark, the authors consider five settings: closed-book, project-structure, location-hint, and two pipelines, considering the last two settings. They find that performance differs by target file type, showing a previously unexplored dimension for fault localisation.

Minor remark: Section 5 uses a potentially old name of the benchmark (PLocBench).

**Strengths:**

- Diversity of issue types and target file types.
- Benchmark is far from saturation under current approaches and a best-effort model for the full dataset.
- Higher difficulty task given the heterogeneity of artefacts that have to be handled.

**Weaknesses:**

- Limited to Python.
- Unclear handling of issues closed with "works as intended" or similar null-reference issues.
- Diverges from how fault localisation is defined in software engineering (this is countered by naming the new setting issue localization).

**Questions:**

- Q1: How are "works as intended" or similar null-reference issues handled? Do you expect a null response from the LLM? Do you expect the implied but not stated reference from documentation on how an API/function should behave? (Please note that I mean null-reference from the issue only; there may be an implied location given the context, for example, what function/API the issue discusses)
- Q2: In software engineering, the line-level fault localisation is usually defined as "within n lines of target". Did you consider such a relaxation to enable line-level performance assessment? If you did, what is the performance at this granularity? (I expect the result to be similar to the function level, but curious, especially if it is different)
- Q3: Per Figure 8, performance in any out-of-project setting degrades severely. Is this a training data availability issue, or do you assume there is a more fundamental reason? Can you speculate on this aspect? (The question is especially due to these new settings being unique to the issue localisation setting, as a generalisation of fault localisation/bug fixing)

---

> ### Author Response · Authors · 2025-11-20
>
> > W1: Limited to Python.
>
> Thanks for your comment! We focus on Python projects for the following reasons:
> * **Practical starting point:** Going beyond code-only bug fix issues is challenging, including data construction methods, manual effort, systematic categorization, and evaluation. Python’s popularity and the focus of popular benchmarks make it a natural initial choice. Importantly, the methodology, categorization, and evaluation framework are language-independent. MULocBench is the first benchmark involving diverse issues and locations, highlighting the complexity of real-world issue localization and providing a foundational testbed for automated data construction and issue localization.
>
> * **Comparability with prior work:** Nearly all state-of-the-art issue localization methods (Agentless, LocAgent) and mature benchmarks (SWE-Bench, LocBench) are limited to Python. Using Python projects allows fair comparisons and reflects the upper bound of current methods’ performance.
> * **Not only Python files:** In Python projects, issue locations are not limited to Python files. Around 24% of issues have locations that do not include any Python files.  We have added statistics on file types for all locations in Appendix A.2.
> | File Type | Issue number | File number |
> |-----------|-------------|------------|
> | Python    | 842         | 2494       |
> | OtherPL (JavaScript/C/Java)  | 16          | 53         |
> | Test  | 273         | 513        |
> | Doc  | 259        | 443        |
> | Asset  | 48        | 59        |
> | Configuration  | 167         | 308        |
>
>
>
>
>
>
>
>
> > W2: Unclear handling of issues closed with "works as intended" or similar null-reference issues.
>
> > Q1: How are "works as intended" or similar null-reference issues handled? Do you expect a null response from the LLM? Do you expect the implied but not stated reference from documentation on how an API/function should behave? (Please note that I mean null-reference from the issue only; there may be an implied location given the context, for example, what function/API the issue discusses)
>
> Thanks for your comment! Following prior work in issue localization (SWE-Bench, LocBench), we focus on real problems that require localization. Therefore, issues closed with “works as intended” (false issues) or null-reference issues (without locations) are excluded. Consistent with prior work (Agentless, LocAgent), we focus on localization ability and do not expect the LLM to produce null responses.
>
> For issues related to documentation (e.g., parameter meaning, usage guide, or document updates), the location contains the document. For non-documentation issues where location involves APIs, we do not include API documentation, as it is easily searchable and can greatly increase location size without benefiting localization tasks. APIs mentioned in the issue description are included only if APIs are necessary for issue resolution; otherwise, APIs are excluded.
>
>
>
>
>
>
> > W3: Diverges from how fault localisation is defined in software engineering (this is countered by naming the new setting issue localization).
>
> Thanks for your comment! Following prior research [1,2], we adopt the term issue localization. Unlike traditional fault localization, which aims to pinpoint the exact faulty code causing code failure, issue localization seeks to identify project locations for issue resolution. Issues are typically expressed in natural language and cover a wide range of intents (e.g., bug reports, feature requests, usage questions). Target locations are not limited to code. This broader scope motivates our task formulation and the choice of the term ‘issue localization.’
>
> [1] Zhonghao Jiang et al. Issue Localization via LLM-Driven Iterative Code Graph Searching. ASE 2025
>
> [2] Revanth Gangi Reddy et al. SWERANK: Software Issue Localization with Code Ranking. Arxiv

---

> ### Author Response · Authors · 2025-11-20
>
> > Q2: In software engineering, the line-level fault localisation is usually defined as "within n lines of target". Did you consider such a relaxation to enable line-level performance assessment? If you did, what is the performance at this granularity? (I expect the result to be similar to the function level, but curious, especially if it is different)
>
> Thanks for your suggestions! Following prior work on line-level fault localisation [1,2,3], we compute Recall@Top-N at the line level, where a prediction is considered correct if any ground-truth line appears within the top-N predicted lines. We evaluate with N = 1 and N = 5, and the extended results have been added in Appendix A.7. For simplicity,  a subset of the results (Table 8) is shown below.
>
> Interestingly, the line-level performance is not similar to the function-level performance, since a target line may not be in code files, and even if it is, it may not be part of any function. For example, the Recall@Top-5 at the line level for Hint-prompt LLMs is 10.8%, which is lower than the Acc@5 at the function level (21.0%).
> | Type | LLM           | File Acc@1 | File Acc@5 | Function Acc@1 | Function Acc@5 | Line Top@1 | Line Top@5 |
> |------|---------------|------------|------------|----------------|----------------|------------|------------|
> | Hint | DeepSeekR1    | 18.0%      | 35.0%      | 16.1%          | 21.0%          | 4.8%       | 10.8%      |
> | Hint | Claude3.5     | 18.6%      | 32.5%      | 9.8%           | 13.1%          | 5.5%       | 6.1%       |
> | Hint | GPT-4o-mini   | 13.5%      | 22.3%      | 6.5%           | 9.9%           | 7.5%       | 9.6%       |
>
>
> [1] Aidan Z.H. Yang et al. Large Language Models for Test-Free Fault Localization ICSE’24
>
> [2] Fang Liu et al. Explainable Fault Localization for Programming Assignments via LLM-Guided Annotation ASE'25
>
> [3] Xia Li et al. DeepFL: Integrating Multiple Fault Diagnosis Dimensions for Deep Fault Localization. ISSTA’19
>
>
>
>
>
>
>
>
>
>
>
> > Q3: Per Figure 8, performance in any out-of-project setting degrades severely. Is this a training data availability issue, or do you assume there is a more fundamental reason? Can you speculate on this aspect? (The question is especially due to these new settings being unique to the issue localisation setting, as a generalisation of fault localisation/bug fixing)
>
> **Limited cross-project reasoning capability.** Thanks for your comment! The degradation in out-of-project shows an intrinsic difficulty: the location lies in another project, yet issue descriptions rarely indicate the responsible project. Although these projects appear in pre-training data, the search space is large, and such cases are less frequent than in-project ones, creating the in-project bias. As a result, LLMs must perform cross-project reasoning, but current LLMs tend to rely on word matching rather than tracing dependencies across loosely coupled codebases. For example, “torch.cuda…” issue is predicted to be torch/torchvision, while the correct project is InterHand2.6M. The details and examples have been included in Appendix A.10.

---

> ### Comment · Reviewer_5shn · 2025-11-21
>
> Thank you for your clarifications and answers, as well as for expanding on the cross-project setting in A.10.
>
> Regarding Q1, I guess my disagreement is with how the literature/work in benchmarking SWE use cases of LLMs does not address the cases where an LLM should remain silent/admit a non-answer. But I am happy to leave this as outside the scope of this work.
>
> Regarding Q2, thank you for exploring the line-level setting as well, and the result is indeed surprising and diverges from what I expected. The explanation seems reasonable for why it diverges, and I do not have any "easy/trivial" extension of that metric to non-code artefacts.
>
> To clarify on W1, since I was perhaps too terse in the weakness to the point that I was (strictly speaking) wrong. The data is primarily from Python projects, but regarding the artefacts themselves, I am happy to accept that they may be of other types. This still skews the data, introducing a bias.
>
> Overall, I maintain my positive rating, with the main concern being the above data bias.

---

> > ### Author Response · Authors · 2025-11-29
> >
> > Thank you for your positive evaluation and the clarification on W1! The choice is consistent with prior work, as existing benchmarks and methods (e.g., SWE-Bench, LocBench, LocAgent, Agentless) are also based on Python projects, ensuring fair and comparable evaluation. Although MULocBench is Python-based, it is the first benchmark that reflects the realistic and complex nature of issue localization, providing a foundation for the community to study and improve issue localization before extending to other languages.
> >
> > We also acknowledge the value of incorporating projects from other languages. Building on MULocBench, future work can extend to additional programming languages and explore cross-language analyses.

---

### Official Review · Reviewer_MvkM · 2025-11-01

**Soundness:** 2
**Presentation:** 2
**Contribution:** 2
**Rating:** 4
**Confidence:** 4

**Summary:**

The paper introduces MULocBench, a comprehensive dataset designed to advance the evaluation of software issue localization methods by addressing the limitations of existing benchmarks. The dataset includes a diverse range of issue types, root causes, location scopes, and file types, encompassing non-code files such as commits, comments, configurations, and documentation. An evaluation of current localization methods and LLM-based prompting strategies reveals significant limitations, with performance metrics remaining below 40% even at the file level. The authors have made MULocBench publicly available to support future research in project localization for issue resolution.

**Strengths:**

1. Introduction of a comprehensive dataset that addresses the limitations of existing benchmarks by including a diverse range of issue types, root causes, location scopes, and file types.
2. Inclusion of non-code files such as commits, comments, configurations, and documentation, providing a more realistic testbed for evaluating localization methods.
3. Public release of MULocBench, facilitating future research and development in project localization for issue resolution, with a close perspective to practical software maintenance.

**Weaknesses:**

1. The dataset consists solely of issues from GitHub Python projects, which may limit its applicability to other programming languages or platforms.
2. Performance metrics (Acc@5, F1) remain below 40% even at the file level, indicating that current methods may not be effective in real-world scenarios.
3. The paper does not provide detailed information on the specific characteristics of the issues included in MULocBench, such as their complexity or the nature of the root causes.

**Questions:**

1. What criteria were used to select the 46 GitHub Python projects included in MULocBench?
2. What specific characteristics of the issues in MULocBench contribute to the challenges in localization, and how can these be mitigated?
3. How do the authors plan to address the identified limitations in current localization methods in future work?

---

> ### Author Response · Authors · 2025-11-20
>
> > W1: The dataset consists solely of issues from GitHub Python projects, which may limit its applicability to other programming languages or platforms.
>
> Thanks for your comment! We focus on GitHub Python projects for the following reasons:
> * **Practical starting point & generality:** Going beyond code-only bug fix issues is challenging, including data construction methods, manual effort, systematic categorization, and evaluation. The popularity of Python and GitHub, and their prominence in existing benchmarks, make Python a natural starting point.
>
>
>      Although the dataset focuses on Python projects, it provides the first benchmark for realistic and complex issue localization and establishes a foundation for the community to study, compare, and improve methods before extending to other programming languages. Importantly, the methodology, categorization, and evaluation framework are language- and platform-independent. Prior work also shows strong cross-language correlations [1, 2], suggesting that clarifying the diversity and complexity of issues and locations is more beneficial at this stage than simply adding additional languages and platforms.
>
>  * **Comparability with prior work:** Nearly all state-of-the-art issue localization methods (Agentless, LocAgent) and mature benchmarks (SWE-Bench, LocBench) are limited to Python. Using Python projects allows fair comparisons and reflects current upper-bound performance.
> * **Not only Python files:** Although the dataset is from Python projects, issue locations are not restricted to Python files. Around 24% of issues have locations that do not include any Python files.  We have added statistics on file types for all locations in Appendix A.2.
> | File Type | Issue number | File number |
> |-----------|-------------|------------|
> | Python    | 842         | 2494       |
> | OtherPL (JavaScript/C/Java)  | 16          | 53         |
> | Test  | 273         | 513        |
> | Doc  | 259        | 443        |
> | Asset  | 48        | 59        |
> | Configuration  | 167         | 308        |
>
> [1] Rozière, Baptiste, et al. Code Llama: Open Foundation Models for Code. arXiv.
>
> [2] Cassano, Federico, et al. MultiPL-E: A Scalable and Extensible Approach to Benchmarking Neural Code Generation. arXiv
>
> > W2: Performance metrics (Acc@5, F1) remain below 40% even at the file level, indicating that current methods may not be effective in real-world scenarios.
>
> Thanks for your comment! We would like to clarify that the low performance of current methods on MULocBench **is not** a weakness, but rather a contribution of benchmarking research, as seen in well-known benchmarks such as GLUE, HumanEval, and BigCodeBench. MULocBench reveals the limitations of existing methods, points out the insufficiency of prior benchmarks that focus only on code/bug-fix/pull-request tasks, reflect the diversity of realistic issues, and offers a testbed to drive the development of new techniques.
>
>
> > Q1: What criteria were used to select the 46 GitHub Python projects included in MULocBench?
>
> Thanks for your comment!  We selected the top-50 most-starred GitHub Python projects and randomly sampled 200 issues from each project. After applying the filtering criteria (keeping only resolved issues with clear locations), the resulting dataset contains 1,100 issues from 46 projects. Details can be found in Section 2, and the project names are provided in the Appendix A.2 to illustrate the diversity of projects included in MULocBench.

---

> ### Author Response · Authors · 2025-11-20
>
> > W3: The paper does not provide detailed information on the specific characteristics of the issues included in MULocBench, such as their complexity or the nature of the root causes.
>
> > Q2: What specific characteristics of the issues in MULocBench contribute to the challenges in localization, and how can these be mitigated?
>
> **Issue Characteristics Analysis** Thanks for your suggestion! We analyze nine issue characteristics in MULocBench, detailed in Appendix A.3. The main findings are as follows:
> * The values of issue characteristics exhibit high variability, reflecting the diverse nature of issues and their potential impact on localization difficulty.
> * Location-related features, especially the number of files and different file types, consistently show strong negative correlations across all LLMs, indicating that **issues involving more extensive locations are generally harder to localize**.
>
> * Textual features show weaker and more variable correlations. Description length has only a weak correlation (<0.01), while richer textual information (e.g., tracebacks) provides modest correlation overall. However, the effect of richer textual features varies across LLMs: Pipeline-Hint applied on GPT-4o-mini and Claude 3.5 benefits from tracebacks, whereas DeepSeekR1 is disrupted by tracebacks, resulting in negative correlations. These variable patterns show that, **Although richer textual information can aid localization, current LLMs cannot fully leverage these signals across complex issue contexts, which limits their effectiveness.**
>
>
> | Feature             | Mean     | Variance    | Min   | Max    |
> |--------------------|---------|------------|-------|--------|
> | iss_descr_length     | 615.45  | 2,692,121.36 | 4     | 37,442 |
> | has_traceback        | 0.21    | 0.17       | 0     | 1      |
> | has_code_block       | 0.50    | 0.25       | 0     | 1      |
> | has_inline_code      | 0.64    | 0.23       | 0     | 1      |
> | file_num             | 2.85    | 16.93      | 1     | 30     |
> | cls_num              | 2.11    | 35.48      | 0     | 68     |
> | func_num             | 2.80    | 56.46      | 0     | 114    |
> | line_num             | 22.09   | 8,511.70   | 0     | 1,291  |
> | diff_file_type_num   | 1.49    | 0.58       | 1     | 5      |
>
>
> Note: ‘*’ means p-value<0.05, representing statistically significant correlations between the issue characteristics and localization performance.
> Excerpt of Correlation Analysis for Pipeline-Hint Approach
> | Approach                  | iss_des_length | has_traceback | has_code_block | has_inline_code | file_num | cls_num | func_num | line_num | diff_file_num |
> |---------------------------|----------------|---------------|----------------|-----------------|---------|---------|---------|----------|---------------|
> | Pipeline-Hint GPT-4o-mini | 0.0001*        | 0.2315*       | -0.2248*       | -0.0471*        | -1.7547* | -0.224* | -0.2239* | -0.0188* | -2.9143*      |
> | Pipeline-Hint Claude3.5   | 0.0003*        | 0.1821*       | 0.2549*        | 0.286*          | -2.1216* | -0.2834* | -0.2907* | -0.026*  | -4.3645*      |
> | Pipeline-Hint DeepSeekR1  | 0.0002*        | -0.0841*      | -0.1206*       | 0.2642*         | -1.4023* | -0.1211* | -0.1736* | -0.0125* | -1.5113*      |
>
>
>
>
>
>
>
>
> > Q3: How do the authors plan to address the identified limitations in current localization methods in future work?
>
> Thanks for your comment! The quantitative analysis in response to RQ2 highlights that LLMs cannot effectively leverage information of issue description. To propose plans, we further examined failed issues and identified three main causes. Details and specific examples are provided in Section 7 and Appendix A.11:
> * **Word-Matching Misleading:** LLMs are misled by unrelated keyword overlaps with software file, class, or function names.
>
>
> * **Code-Centric Bias:** LLMs tend to assume issues are code-related, leading to incorrect localization.
>
>
> * **Responsibility Chain Tracking Limitation:** LLMs fail to trace multi-step responsibility paths in issues.
>
> Based on the findings, we plan to address the limitations as follows:
>
> * **Unrelated keywords masking.** Masking words reduces spurious lexical overlaps between issue descriptions and project locations(e.g., file names) so that LLMs can rely more on semantic cues rather than shallow word matching.
> * **File-type–aware keyword mapping.** Construct mapping libraries that associate each file type with representative keywords.These mappings can be used as a part of prompt to alleviate code-centric bias and guide models toward non-code or mixed-context issues.
> * **Project-level responsibility chains construction** Construct project-level responsibility chains that explicitly encode relationships among entities (project/file/class/function) to guide LLMs to perform deterministic deep reasoning.

---

### Author Response · Authors · 2025-11-21
**General Response**

Thanks for all reviewers’  insightful comments. We have incorporated the rebuttal content into our revision to answer the reviewers’ questions.

MULocBench aims to answer two fundamental questions: (1) What do real-world software issues actually look like? and (2) How well do current methods perform on such realistic issue-localization tasks? The contribution is as follows:


* **Multi-Source Real-World Issues**. We gather 1,100 issues from 46 Python projects from multiple sources including pull requests, commits, and comments, rather than relying solely on pull requests.

* **Systematic Taxonomy of Issue Localization**. We define a taxonomy for issue localization across four key dimensions: issue types, root causes, file scopes, and file types, to provide a comprehensive and systematic understanding of real-world issues.

* **Comparative Analysis with Benchmarks**. Unlike previous benchmarks that focus on single-file Python bug fixes within project boundaries, MULocBench covers diverse issue types and causes, spans multiple files, includes both code and non-code artifacts, and extends to locations beyond in-project files. This breadth reflects the true complexity and diversity of real-world software issues.

* **Systematic Evaluation and Analysis**. State-of-the-art localization methods and prompting-based LLMs exhibit consistently poor performance on MULocBench. The results show that issue localization remains far from saturated and highlights the need to improve AI’s deep reasoning for handling complex software issues.


To address reviewers’ suggestions, we have made the following modifications:

* **Failure Analysis**. We have added failure analysis and directions to address the limitations in Section 7 and included five representative failed cases in Appendix A.11.

*  **Extended Experiments**. We have added line-level evaluation experiment in Appendix A.7, temporal analysis in Appendix A.8, expanded cross-project analysis in Appendix A.10, and evaluation on DeepSeek-R1 in Sections 5 and 6.


*  **Clarifications and Emphasis**. We have clarified the selection of the Python project, the distinction between issue localization and fault localization, and data construction details.

---

### Note · Authors · 2026-01-26

I have read and agree with the venue's withdrawal policy on behalf of myself and my co-authors.

---

### Meta-Review · Area_Chair_yATC · 2025-12-31

**Summary:**

This paper introduces MULocBench, a new benchmark for software issue localization. The authors argue that existing benchmarks, such as SWE-Bench and LocBench, are too narrowly focused on bug fixes resolved via pull requests. To address this limitation, they have constructed MULocBench, a diverse dataset of 1100 issues from 46 popular Python projects. The paper then reports on an extensive empirical study evaluating both state-of-the-art specialized tools and several LLM-based prompting strategies on this new benchmark.

**Reviewer Concerns:**

1. The diversity of the data set is limited (e.g., only on GitHub and Python);
2. Lack of details about the dataset and deep analysis;
3. The low performance scores compellingly demonstrate that current models fail and the pape has not provided reasons.

**Reviewer Scores:**

The paper received mixed ratings. After the rebuttal, some of the reviewers' concerns have been addressed, but most of them still hold. AC made the rejection decision.

---

### Decision · Program_Chairs · 2026-01-26

Reject